# SMAFace: Sample Mining Guided Adaptive Loss for Face Recognition

## Abstract

Traditional face recognition (FR) algorithms often rely merely on margin-based softmax loss functions. However, due to varied image hardness in datasets, these models often falter when dealing with low-quality images. To address this issue, we introduce SMAFace, an innovative FR algorithm that enhances performance by incorporating sample mining into conventional margin-based methods. At its core, SMAFace focuses on prioritizing information-dense samples, namely hard samples or easy samples, which present more distinctive features. In this study, we employ a probability-driven mining strategy, enabling the model to adeptly navigate hard samples, thereby bolstering its robustness and adaptability. The mathematical evaluation and empirical tests of SMAFace indicate its effectiveness. Moreover, experimental results reveal that our approach surpasses the state-of-the-art (SoTA) on four renowned datasets (CPLFW, VGG2-FP, IJB-B and Tiny-Face), highlighting its potential and efficiency.

## 1 Introduction

FR constitutes a pivotal task within the realm of computer vision. Metric learning is an effective approach for face recognition and forms the foundation of cosine similarity methods(Sun et al., 2014; Taigman et al., 2014). The quality of face images largely depends on various factors such as brightness, contrast, clarity and noise. For high-quality face images, despite some exhibiting variations in illumination, pose and expression, the task of FR can still be accomplished. However, when faced with low-quality images, such as those impaired by noise or low resolution, the recognition task becomes quite challenging. Traditional FR algorithms usually adopt a margin-based softmax loss function for classification merely, but their performance in dealing with low-quality images is subpar. This is caused by noise and other disturbance factors, which make it difficult for the model to acquire useful features.

In this context, we propose a novel FR algorithm that considers both margin function and sample mining to improve the performance of FR models. The initial step in this direction is to find an effective proxy of image quality (Terhörst et al., 2020; Boutros et al., 2023; Long Chai et al., 2023). We use the feature norm as an approximate proxy of image quality, thereby allowing for an accurate assessment of image quality during the training process and enabling subsequent model optimization based on this.

Sample mining is an effective strategy for enhancing the performance of deep neural networks. This strategy focuses on samples that are rich in information, as they provide more effective discriminatory features. In this paper, we introduce a mining-based strategy by probability to help the model better adapt to hard samples, thereby enhancing the model's robustness and generalization capabilities (see Figure 1).

We define a parameter called the scaling term. Its fundamental source lies in the partial derivatives of the backpropagation process, namely stochastic gradient descent (SGD). This is not an unfamiliar concept; rather, it is a crucial and widely acknowledged element. We subjected it to a brief mathematical analysis, subsequently demonstrating that the handling perspective of hard and easy samples can indeed yield the intended effects. We also engaged in visualizing its operations, enabling us to readily discern the underlying concept of SMAFace.

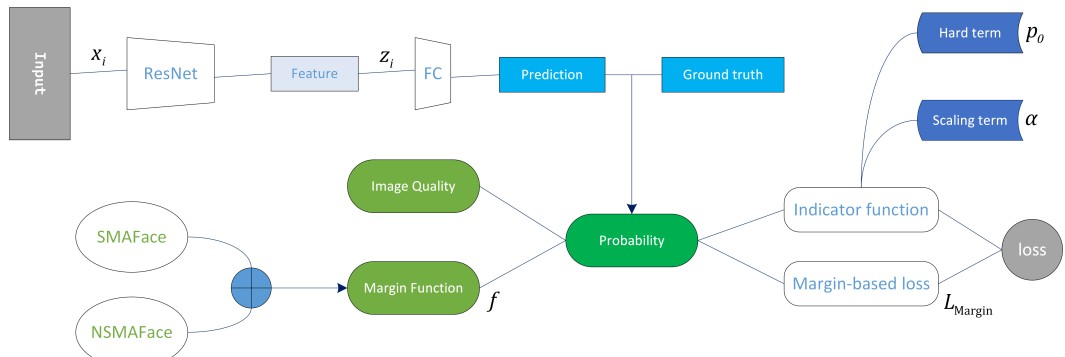

Figure 1: FR based-mining algorithm by probability.

To validate the effectiveness of our proposed new loss function and mining strategy, we performed experiments on multiple datasets and compared our method with other approaches from recent years. The experimental results indicate that our method has achieved significant performance improvements, demonstrating excellent efficacy and potential.

In summary, our main contributions include:

- We introduce a mining-based loss function by probability to enhance the model's robustness and generalization capabilities.
- We propose a parameter referred to as the scaling term to analyze the nature of the model, derived from a series of mathematical derivations to serve as a metric of measurement.
- We visualize its values through heatmap representations, which facilitate a more effective understanding of the differences among various methods.
- We validate the effectiveness of our method through experiments on multiple datasets. The results show that our method has a significant performance advantage in the task of FR.

## 2 RELATED WORK

**Margin-based Loss Function**  The cross-entropy loss for sample $\boldsymbol{x}_i$ can be represented as

$$\mathcal{L}_{CE}(\boldsymbol{x}_i) = -\log \frac{\exp(\boldsymbol{W}_{:,y_i}\boldsymbol{z}_i + b_{y_i})}{\sum_{j=1}^{C}\exp(\boldsymbol{W}_{:,j}\boldsymbol{z}_j + b_j)},$$

where $y_i$ stands for the index of the ground truth(GT) label and $\boldsymbol{z}_i \in \mathbb{R}^d$ is the feature input for $\boldsymbol{x}_i$ which belongs to the $y_i^{\text{th}}$ class. $\boldsymbol{W}_{:,j}$ denotes the $j^{\text{th}}$ column of the final fully connected layer weight matrix, $\boldsymbol{W} \in \mathbb{R}^{d \times C}$, and $b_j$ represents the corresponding bias term, with $C$ denoting the number of classes.

The introduction of cosine similarity has revolutionized FR approaches. To make classification more effective and in line with the characteristics of FR, Liu et al. (2017); Wang et al. (2017) employed a normalized softmax method, setting the bias term to 0. During training, they normalized $\boldsymbol{z}_i$ and scaled it with $s$. The modified formulation becomes

$$\mathcal{L}_{CE}(\boldsymbol{x}_i) = -\log \frac{\exp(s \cdot \cos\theta_{y_i})}{\sum_{j=1}^{C}\exp(s\cos\theta_j)},$$

where $\theta_j$ is the angle between $\boldsymbol{z}_i$ and $\boldsymbol{W}_{:,j}$. Subsequent research (Deng et al., 2019a; Wang et al., 2018; Liu et al., 2017) adopted this approach and introduced a margin to decrease the intra-class distance. Generally, it can be expressed as

$$\mathcal{L}_{CE}(\boldsymbol{x}_i) = -\log \frac{\exp(f(\theta_{y_i}; m))}{\exp(f(\theta_{y_i}; m)) + \sum_{j \neq y_i}^{C}\exp(s\cos\theta_j)}, \tag{1}$$

where $f(\theta_{y_i}; m)$ denotes the margin function and $m$ is a constant hyperparameter representing the margin. SphereFace(Liu et al., 2017), CosFace(Wang et al., 2018) and ArcFace(Deng et al., 2019a) each proposed a unique margin function to enhance clarity in feature space discrimination. In the equations mentioned above, SphereFace employs the function $f(\theta_{y_i}; m) = s\cos(m\theta_{y_i})$, CosFace utilizes the function $f(\theta_{y_i}; m) = s(\cos(\theta_{y_i}) - m)$, while ArcFace adopts the function $f(\theta_{y_i}; m) = s\cos(\theta_{y_i} + m)$. The design of these formulas fully considers the distance between classes and the distribution of features, enabling the model to better recognize and differentiate faces.

**Adaptive Loss Functions**   The design of adaptive loss functions, due to their characteristics of self-adjustment according to the process and effect of learning, brings new possibilities for the training and optimization of deep learning models. Among them, the proposal of the AdaCos(Zhang et al., 2019) resolves the cumbersome and inefficient process of parameter tuning. The AdaCos builds upon the CosFace approach and further introduces a new method that automatically modifies the margin, and scales hyperparameters $m$ and $s$ values with training iterations. AdaCos observes that when the $s$ value is too small, even though the angular margin $\theta(i, y_i)$ between target $i$ and its class center is sufficiently small, the probability of it being classified as a positive instance is relatively low. Conversely, when the $s$ value is too large, even if the angular margin $\theta(i, y_i)$ between target $i$ and its class center is still relatively large, its probability of being classified as a positive instance may approach 1. Based on some theoretical derivations, AdaCos proposes a method for updating adaptive parameters from a mathematical perspective.

AdaptiveFace(Liu et al., 2019) has proposed an AdaM-Softmax loss, which is modified based on CosFace. As training progresses, AdaM-Softmax will give larger margins to poor classes and smaller margins to rich classes. This strategy fully considers the characteristics of the training set and helps improve the overall performance of the model.

CurricularFace(Huang et al., 2020b) is also an adaptive loss function, which gradually increases in difficulty as the training progresses. Its concept is easier to comprehend than the previous two approaches, encapsulated in a single sentence. However, it is similarly ingeniously constructed and its solidness is on par with them.

**Loss Functions for Low-Quality Images**   In the handling of loss functions for low-quality images, MagFace(Meng et al., 2021) has introduced an adaptive mechanism that draws relatively easy samples closer to class centers of a larger magnitude while pushing more hard samples away from the center, reducing their magnitude in the process. This mechanism enables the learning of intra-class feature distribution with a good structure. Specifically, the loss function of MagFace is expressed as follows

$$\mathcal{L}_{MagFace} = -\log \frac{e^{s\cos(\theta_{y_i} + m(a_i))}}{e^{s\cos(\theta_{y_i} + m(a_i))} + \sum_{j \neq y_i} e^{s\cos(\theta_j)}} + \lambda_g g(a_i).$$

In this equation, $a_i$ represents the scale of the non-normalized facial features of samples $i$, $m(a_i)$ denotes the scale-aware angular margin for the positive samples $i$, which is monotonically increasing, and $g(a_i)$ is a regularizer designed as a monotonically increasing convex function. The parameters $m(a_i)$ and $g(a_i)$ jointly emphasize the orientation and scale of face embedding, and $\lambda_g$ is a parameter used to balance these two parameters.

Although MagFace introduces the concept of scale, it does not further introduce the concept of feature norms. Feature norms, bearing many similarities to scale, were first proposed for FR in AdaFace(Kim et al., 2022). MagFace emphasizes easy samples for higher quality samples, as it aims to place high-quality easy samples in the central region and both low-quality easy samples and high-quality hard samples in the peripheral region. Conversely, AdaFace values high-quality samples of any difficulty.

To address the common issue of data uncertainty in FR, probabilistic face representation learning is one way to solve this problem, with the latest method being Spherical Confidence Learning (Li et al., 2021), also known as SCF-ArcFace. It theoretically demonstrates that concentration values can be interpreted as measures of confidence, and it does not require independent Gaussian assumptions or paired training. Compared to the maximization of the expected mutual likelihood score in PFE(Shi & Jain, 2019), this framework minimizes the KL divergence between the spherical Dirac delta and $r$-radius vMF, which is then translated into the minimization of cross-entropy. This method takes

advantage of the properties of the Dirac delta function to transform the loss function, thereby forming the latent prior expectation on the sphere and effectively introducing probability to FR.

Improving from misclassified samples (Wang et al., 2020) is a viable approach. Notably, distillation techniques have also been used to increase FR accuracy (Huang et al., 2020a; Li et al., 2023; Huang et al., 2022). BioNet(Li, 2023) significantly improved the accuracy of low-quality datasets using neuroscientific methods. Studies (Feng et al., 2018; Shi et al., 2020; He et al., 2022) involving physical 3D reconstruction ignore identity-irrelevant information. A sample-level weighting approach called MvCoM was proposed from a mathematical perspective to handle various bias changes in Liu et al. (2022). SphereFace2(Wen et al., 2022) circumvents the softmax normalization and bridges the gap between training and evaluation.

## 3 SMAFACE

### 3.1 MINING-BASED SOFTMAX AND ADAPTIVE MARGIN FUNCTION

A pivotal advancement in FR technology is the mining-based strategy of hard samples. The core idea of this strategy revolves around giving special attention and training to so-called hard samples, which can substantially enhance the performance of FR systems in practical applications. The mining-based technique of samples has been gradually acknowledged as an effective method for training deep neural networks. Numerous recent research papers, such as Lin et al. (2017); Shrivastava et al. (2016), have opted to learn discriminative features based on the loss value of hard samples.

In these studies, the mining-based formula can be summarized as

$$\mathcal{L}_{\text{Mining}}(\boldsymbol{x}_i) = -I\left(p_{y_i}\right) \log \frac{\exp(f(\theta_{y_i}; m))}{\exp(f(\theta_{y_i}; m)) + \sum_{j \neq y_i}^{C} \exp(s \cos \theta_j)}, \tag{2}$$

where, $p_{y_i} = \frac{\exp(f(\theta_{y_i}; m))}{\exp(f(\theta_{y_i}; m)) + \sum_{j \neq y_i}^{C} \exp(s \cos \theta_j)}$ represents the probability of the prediction being the ground truth and $I(p_{y_i})$ is an indicator function. Drawing upon the research in Lin et al. (2017), they introduced F-Softmax, while Shrivastava et al. (2016) presented HM-Softmax. The specific forms are as $I_{\text{HM-Softmax}}(p_{y_i}) = \begin{cases} 0 & \textit{the sample is easy} \\ 1 & \textit{the sample is hard} \end{cases}$ and $I_{\text{F-Softmax}}(p_{y_i}) = (1 - p_{y_i})^{\gamma}$. The $\gamma$ is a modulating factor.

However, we have proposed a new definition for the indicator function, which differs from the methods proposed by Lin et al. (2017); Shrivastava et al. (2016). Our definition is given by

$$I(p_{y_i}) = 1 + \alpha \left( \frac{1}{1 + e^{p_{y_i} - p_0}} - 0.5 \right), \tag{3}$$

In this equation, $\alpha$ is a key hyperparameter, primarily used to control the magnitude of the weights. First, let's consider the case where $\alpha$ is a positive number. When a sample is harder, meaning that $p_{y_i}$ is smaller, the weight $I(p_{y_i})$ becomes larger. Conversely, when the sample is easier, with $p_{y_i}$ being larger, the weight $I(p_{y_i})$ becomes smaller. Moreover, $p_0$ acts as a threshold, primarily adjusting the boundary at which a sample is considered hard or easy. If the $p_{y_i}$ of a sample is less than $p_0$, the sample is viewed as hard, and the weight $I(p_{y_i})$ will be greater than 1. In contrast, if the $p_{y_i}$ of a sample is greater than $p_0$, the sample is considered easy, and the weight $I(p_{y_i})$ will be less than 1. For the case where $\alpha$ is a negative value, the situation is opposite to what was described earlier. During experiments, we found that this indicator function results in $loss = NaN$ for non-conventional values of $\alpha$. Thus, for cases with $p_0 = 0.6$, we reset $I^*(p_{y_i}) = \left\| \frac{7}{7+\alpha} \right\| \left( 1 + \alpha \left( \frac{1}{1 + e^{p_{y_i} - p_0}} - 0.5 \right) \right)$, where the factor of 7 is solely related to $p_0$ and is an empirically optimal value..

Next, by incorporating this dynamic weighting coefficient into Equation 2, we can obtain our final loss function. It's worth noting that, unlike F-Softmax and HM-Softmax, this new weighting function offers a more valuable distinction between easy and hard samples, as we introduce a dynamically adjusted weight coefficient based on the correct class probability in the loss function. Such a design allows the model to focus more on samples that we prefer during training, thereby enhancing its discriminative capabilities.

In practical applications, it's necessary to adjust the hyperparameter $\alpha$ and threshold probability $p_0$ based on the characteristics of the dataset and task requirements. These parameters will influence the calculation of the weight coefficient $I(p_{y_i})$, subsequently affecting the model's attention towards samples of varying difficulties. By judiciously setting these parameters, we can balance the weights of easy and hard samples during training, thereby effectively boosting the model's performance in FR tasks.

Regarding the adaptive margin function, we define it as

$$f(\theta_j; m)_{\text{SM-CosFace}} = \begin{cases} s(\cos\theta_j - m_{\text{add}}) & j = y_i \\ s\cos\theta_j & j \neq y_i \end{cases}, \tag{4}$$

$$f(\theta_j; m)_{\text{SMAFace}} = \begin{cases} s(\cos(\theta_j + m_{\text{angle}}) - m_{\text{add}}) & j = y_i \\ s\cos\theta_j & j \neq y_i \end{cases}. \tag{5}$$

Equations 4 and 5 are the margin functions we will employ. SM-CosFace is used to demonstrate the positive effects of sample mining when compared to CosFace, while SMAFace will be employed for comparisons with other methods.

## 3.2 SCALING TERM FROM GST

At the outset of this section, we employ the definition of the gradient scaling parameter (GST)

$$g := \frac{\partial \mathcal{L}_{\text{Margin}}}{\partial p_j^{(i)}} \frac{\partial p_j^{(i)}}{\partial f(\cos\theta_j)} \frac{\partial f(\cos\theta_j)}{\partial \cos\theta_j}. \tag{6}$$

$g_{\text{Mining}}$ is decomposed into $g_{\text{SM-CosFace}}$ and $g_{\text{SMAFace}}$. The value of $g_{\text{SM-CosFace}}$ is

$$g_{\text{SM-CosFace}} = \left(\frac{\partial I(p_{y_i})}{\partial p_{y_i}} p_{y_i} \log p_{y_i} + I(p_{y_i})\right)(p_{y_i} - 1)s, \frac{\partial I(p_{y_i})}{\partial p_{y_i}} = \alpha\sigma(p_0 - p_{y_i})(\sigma(p_0 - p_{y_i}) - 1), \tag{7}$$

where $\sigma(x)$ denotes the sigmoid function. For the proofs of Equation 7, see Appendix C and D. The proof processes for these are not particularly important, and simply examining the computational results will not impact the understanding of this paper.

Given that it signifies a discrepancy, the magnitude's significance increases proportionally. By comparing $g_{\text{softmax}}$, $g_{\text{CosFace}}$, $g_{\text{ArcFace}}$ and $g_{\text{SM-CosFace}}$, we note that $(p_{y_i} - 1)s$ is a common term across these equations. Therefore, we can normalize these equations by the base value $(p_{y_i} - 1)s$, renaming the result scaling term ($st$), to highlight their difference. It can be observed that $st_{\text{softmax}} = 1$ and $st_{\text{CosFace}} = 1$, while

$$st_{\text{ArcFace}} = \cos(m) + \frac{\cos\theta_{y_i}\sin(m)}{\sqrt{1 - \cos^2\theta_{y_i}}}, st_{\text{SM-CosFace}} = \frac{\partial I(p_{y_i})}{\partial p_{y_i}} p_{y_i} \log p_{y_i} + I(p_{y_i}). \tag{8}$$

There are specific reasons behind both scaling parameters not being equal to 1, and these reasons differ. The reason why $st_{\text{ArcFace}}$ is not equal to 1 is because the result of $\frac{\partial f(\cos\theta_{y_i})}{\partial \cos\theta_{y_i}}$ is not $s$. Conversely, the reason why $st_{\text{SM-CosFace}}$ is not 1 comes from the fact that the result of $\frac{\partial \mathcal{L}}{\partial p_{y_i}}$ is not $-\frac{1}{p_{y_i}}$. This leads to the dynamic adaptive nature of $st$. As these origins are different, they can be used in conjunction.

We have plotted the variations of $st$ with different $p_{y_i}$ values for $p_0$ and $\alpha$, and compared the $st$ values between ArcFace and SM-CosFace. It can be observed that SM-CosFace demonstrates better balance and controllability when dealing with samples of varying difficulties. The performance under various scenarios can be flexibly modulated by adjusting the parameter $\alpha$.

From the perspective of $st$, the principles behind AdaFace and ArcFace are congruent. Thus, as discussed previously, the method of AdaFace can be amalgamated with SM-CosFace, culminating in the creation of SMAFace. The $st_{\text{SMAFace}}$ is given by

$$st_{\text{SMAFace}} = \left(\cos(m_{\text{angle}}) + \frac{\cos\theta_{y_i}\sin(m_{\text{angle}})}{\sqrt{1 - \cos^2\theta_{y_i}}}\right)st_{\text{SM-CosFace}}. \tag{9}$$

From this point, the derivation of $g_{\text{SMAFace}}$ becomes evident, and the analysis for $g_{\text{Mining}}$ that we propose is concluded.

Earlier, we introduced the indicator function $I^*(p_{y_i})$. The results obtained from this method are termed SMAFace*. It results in distinct $st$ values. Notably, when $\alpha$ takes on negative values, the dynamics shift profoundly, which refers to NSMAFace. It's imperative to ensure $\alpha > -7$. The properties of SMAFace* and SMAFace remain consistent, so no distinction will be made in subsequent discussions. The same applies to NSMAFace* and NSMAFace.

### 3.3 Comparison and Analysis

In Figure 2, the vertical axis $\widehat{\|z_i\|}$ represents a proxy for image quality. We can find that when $\widehat{\|z_i\|} = -1$, the corresponding $m_{\text{angle}}$ is $m$, and here we take it as $0.4$. This value corresponds to an image with a low feature norm, implying low quality, and the converse holds. Examining AdaFace's performance, it indeed prioritizes easy samples when facing low-quality images and hard samples when dealing with high-quality images. In our method, when $\alpha > 0$ and is no larger than 7, we do emphasize hard samples when facing low-quality images, but we don't overlook easy samples, implementing a bimodal strategy. When dealing with high-quality images, our attention to hard samples surpasses that of AdaFace. When $\alpha < 0$ and no smaller than $-7$, we also adopt a bimodal strategy for high-quality images, but our focus on easy samples is even more intense than AdaFace. Overall, SMAFace demonstrates a heightened focus on hard samples, whereas NSMAFace directs its attention predominantly to easy samples.

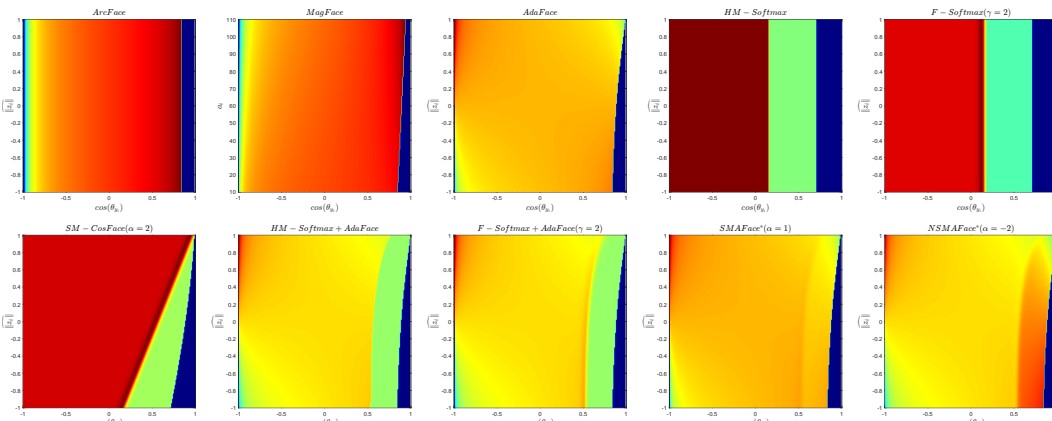

Figure 2: Comparison of ten method concerning $\widehat{\|z_i\|}$ and $cos(\theta_{y_i})$. MagFace depends on $a_i$ and $cos(\theta_{y_i})$, where $a_i$ represents the magnitude, similar to $\widehat{\|z_i\|}$.

We present the heatmap of the scaling term for ten scenarios in Figure 2. The conversion of $st$ into color values is not merely a straightforward process; rather, it involves a consideration of the angle margin and cosine margin. The blue region on the right-hand side represents cases where the inequality $f(\theta_{y_i}; m) > s \cos \theta_j (j \neq y_i)$ is satisfied. In such instances, correct classification can be determined without the need for any penalty. Herein, a hotter hue indicates a larger value while a colder hue signifies a smaller one. As $\widehat{\|z_i\|}$ increases, the image quality improves. Furthermore, larger $cos(\theta_{y_i})$ values suggest that the image is easier to recognize. For HM-Softmax, F-Softmax and their combination with AdaFace, there exists an issue where the $st$ value of the green region on the right is set to $0$. This is unreasonable, as we shouldn't completely disregard these easy samples. In other words, both of these Mining-based Methods have this irreparable issue, which is why we propose SMAFace.

Notably, within the plots for SM-CosFace, SMAFace* and NSMAFace*, a linear boundary is observed. This boundary exhibits a narrow width, attributed to the fact that $s = 64$. As the value of $s$ increases, its width shrinks. The slope of this boundary is determined by $m$, with $m = 0.4$ in this context, leading the boundary to pass through the point $(0.4, -1)$. This division positions easy samples towards the right region and hard samples towards the left. We observe that SMAFace*

places a higher emphasis on harder samples, especially those of higher quality, thus aiding hard and high-quality samples in garnering the attention they deserve. On the other hand, NSMAFace* prioritizes easier samples, particularly those of lower quality, ensuring they receive appropriate attention. Deciding whether hard or easy samples contribute more substantially to the results remains a context-dependent consideration.

## 4 EXPERIMENTS AND RESULTS

### 4.1 DATASETS AND IMPLEMENTATION DETAILS

**Datasets**    In our research, we utilize three training datasets, namely MS1MV2(Deng et al., 2019a), MS1MV3(Deng et al., 2019b) and WebFace4M(Zhu et al., 2021), which provide a rich set of training samples for our study. Our evaluations are performed on ten distinct datasets to validate the model's performance.

Among these datasets, LFW(Huang et al., 2008), CFP-FP(Sengupta et al., 2016), CPLFW(Zheng & Deng, 2018), AgeDB(Moschoglou et al., 2017), CALFW(Zheng et al., 2017), CFP-FF(Sengupta et al., 2016) and VGG2-FP(Cao et al., 2018) are highly popular benchmark datasets. The image quality in these datasets is exceptionally high, aiding us in accurately evaluating our model's performance. The IJB-B and IJB-C(Whitelam et al., 2017; Maze et al., 2018) datasets introduce some low-quality images, which help us assess the model's generalization capabilities. Both IJB-B and IJB-C include a mix of high and low-quality face images. Lastly, TinyFace(Cheng et al., 2019) is a pure low-quality dataset without any high-quality face images, proving essential for testing the model's performance under extreme conditions.

**Training Settings**    Throughout our research, we adopted and followed the processing method from Deng et al. (2019a). Initially, face images were finely cropped using MTCNN and aligned to five preset landmarks. By doing so, we generated uniformly sized face images of dimensions $112 \times 112$. For the primary neural network architecture, our backbone network, we referenced and employed the improved version of ResNet(He et al., 2016) from Deng et al. (2019a). On this foundation, several rounds of training were carried out. We trained for a total of 30 epochs, using SGD as our optimization method. Regarding the learning rate setting, we adopted a dynamic adjustment strategy. The initial learning rate was set to $0.1$, and during the training process, we reduced the learning rate by a factor of ten at the $12^{th}$, $20^{th}$, and $24^{th}$ epochs to achieve better training outcomes. For the scaling parameter $s$, we set it to $64$, referencing the standard settings from Deng et al. (2019a); Wang et al. (2018). Regarding data augmentation, we incorporated three widely used augmentation techniques in image classification tasks (He et al., 2019), including cropping, rescaling, and luminance modification. The probability of applying these techniques was uniformly set to $0.2$, identical to that in AdaFace (Kim et al., 2022), to enhance the model's generalization. Lastly, in terms of training batch size, for ResNet18 and ResNet50, we set the batch size to $256$, while for ResNet100, it was set to $512$. Hardware-wise, ResNet18 and ResNet50 were trained on a single-card NVIDIA GeForce RTX 4090 server, whereas ResNet100 was trained on single-card NVIDIA A100 80GB PCIe and five-card NVIDIA RTX A4000 server. The training precision for all was set to $16bit$.

### 4.2 ABLATION AND ANALYSIS

In this study, we conducted an in-depth ablation analysis on the hyperparameters $\alpha$ and $p_0$. To accurately assess the effects of these hyperparameters, we employed ResNet50 and ResNet18 as the backbone networks, utilizing MS1MV2 as our experimental dataset. The performance metrics we adopted include the average 1:1 verification accuracy on LFW, CFP-FP, CPLFW, AgeDB, CALFW, CFP-FF and VGG2-FP.

**Effect of Image Hard Threshold Concentration** $p_0$    In our initial hypothesis, we anticipated the optimal value for hyperparameter $p_0$ might be 0.5. However, to validate this assumption, a series of empirical experiments were necessary. In Table 1, we showcase performance comparisons at various $p_0$ values, concluding that the best value for $p_0$ is $0.6$. This decision primarily stems from the average accuracy of high-quality datasets. Achieving high accuracy on low-quality datasets is

Table 1: Ablation of our mining function parameters $p_0$ and $\alpha$ and the mining function parameters $m$ on the ResNet18 backbone and the ResNet50 backbone. The performance metrics are as described in Section 4.2.

| Method | Backbone | $p_0$ | $\alpha$ | HQ Datasets |
|---|---|---|---|---|
| AdaFace (Reproduce) | ResNet50 | - | - | 97.00 |
| SMAFace | ResNet50 | 0.40
0.50
**0.60**
0.70
0.80 | 1.00 | 97.02
97.02
**97.09**
97.00
96.97 |
| SMAFace
SMAFace* | ResNet50 | 0.6 | 0.50
**1.00**
2.00 | 97.06
**97.09**
96.99 |
| NSMAFace* | ResNet50 | 0.6 | −1.00
**−3.00**
−5.00 | 97.02
**97.04**
96.98 |
| AdaFace (Reproduce) | ResNet18 | - | - | 95.75 |
| SMAFace
SMAFace* | ResNet18 | 0.6 | 0.50
**1.00**
2.00 | 95.69
**95.70**
95.65 |
| NSMAFace* | ResNet18 | 0.6 | −1.00
**−3.00**
−5.00 | 95.76
**95.81**
95.78 |
| CosFace (Reproduce) | ResNet50 | - | - | 96.94 |
| SM-CosFace | ResNet50 | 0.60 | 1.00 | 96.99 |
| NSM-CosFace* | ResNet50 | 0.6 | −1.00
**−3.00**
−5.00 | **97.04**
97.03
96.97 |

contingent upon already attaining it on high-quality ones. Moreover, the accuracy of AdaFace is based on replicated results, using a batch size of 256.

**Effect of Scaling Hyperparameter** $\alpha$    The scaling factor $\alpha$ is a pivotal parameter in our method, and its effects were previously analyzed through heatmaps. When $\alpha > 0$, our method is termed SMAFace; whereas, for $\alpha < 0$, it's called NSMAFace. To prevent scenarios of $loss = NaN$ for $\alpha > 1$ and $\alpha < 0$, we employed the scaled versions SMAFace* and NSMAFace*. This practice aids in model convergence. Table 1 displays experiments to investigate the impact of varying $\alpha$ values. As previously mentioned, different scaling factors modify the model's focus on samples of varying difficulty, subsequently influencing the final results. Larger $\alpha$ values increase the model's attention to hard samples, while smaller values emphasize easy samples. Given MS1MV2's mediocre quality, prioritizing hard samples isn't always advisable. Observing Table 1 reveals commendable results for $\alpha = 1, -3$. Interestingly, due to ResNet18's limited fitting capability, it struggles with hard samples when $\alpha > 0$, yielding worse performance on HQ Datasets compared to AdaFace. The performance of SM-CosFace also demonstrates a significant improvement over CosFace.

## 4.3 COMPARISON WITH SoTA METHODS

To compare with the SoTA methods, we trained the ResNet100 model using both SMAFace and NSMAFace and evaluated it on the 10 datasets mentioned in Section 4.1. For high-quality datasets in Table 2, it can be observed that the accuracies are all above $90\%$, leaving little room for improvement. The accuracy of these datasets is nearing saturation, making breakthroughs challenging. On some easier datasets, such as LFW, AgeDB and CFP-FF, our method did not lead to significant improvements. However, SMAFace, especially in the challenging CPLFW and VGG2-FP under HQ Quality, exhibited impressive performances. Given that SMAFace emphasizes hard samples, this underscores the importance of these hard samples in training. The model trained on WebFace4M

Table 2: A performance comparison of recent methods ArcFace(Deng et al., 2019a), AFRN(Kang et al., 2019), MV-Softmax(Wang et al., 2020), CurricularFace(Huang et al., 2020b), BroadFace(Kim et al., 2020), MagFace(Meng et al., 2021), SCF-ArcFace(Li et al., 2021), MvCoM-CosFace(Liu et al., 2022), AdaFace(Kim et al., 2022), VPL-ArcFace(Deng et al., 2021) and PFC-0.3(An et al., 2022) on high quality datasets with the ResNet100 backbone. For LFW, CFP-FP, CPLFW, AgeDB, CALFW, CFP-FF and VGG2-FP, 1:1 verification accuracy is reported. The first table contains training sets all from MS1MV2, the second table consists of training sets from MS1MV3, and the third one utilizes WebFace4M. All SMAFace variants have $\alpha = 1$, SMAFace* has $\alpha = 2$ and NSMAFace* variants all have $\alpha = -2$. The red data in the table represents the state of the art, while the blue data corresponds to the second-highest accuracy ranking.

| Method | High Quality | | | | | | | |
|---|---|---|---|---|---|---|---|---|
| | LFW | CFP-FP | CPLFW | AgeDB | CALFW | CFP-FF | VGG2-FP | AVG |
| ArcFace | 99.83 | 98.27 | 92.08 | 98.28 | 95.45 | | | |
| AFRN | 99.85 | 95.56 | 93.48 | 95.35 | 96.30 | | | |
| MV-Softmax | 99.80 | 98.28 | 92.83 | 97.95 | 96.10 | | | |
| CurricularFace | 99.80 | 98.37 | 93.13 | 98.32 | 96.20 | | | |
| BroadFace | 99.85 | 98.63 | 93.17 | 98.38 | 96.20 | | | |
| MagFace | 99.83 | 98.46 | 92.87 | 98.17 | 96.15 | | | |
| SCF-ArcFace | 99.82 | 98.40 | 93.16 | 98.30 | 96.12 | | | |
| MvCoM | 99.80 | 98.37 | 92.75 | | | | | |
| AdaFace (Reproduce) | 99.78 | 98.54 | 93.20 | 98.10 | 96.20 | 99.81 | 95.80 | 97.35 |
| **SMAFace** | 99.83 | 98.47 | 93.72 | 98.27 | 96.10 | 99.79 | 95.54 | 97.39 |
| **NSMAFace*** | 99.82 | 98.56 | 93.45 | 98.32 | 96.12 | 99.79 | 95.48 | 97.36 |
| VPL-ArcFace | 99.83 | 99.11 | 93.45 | 98.60 | 96.12 | | | |
| AdaFace (Reproduce) | 99.82 | 98.97 | 93.67 | 98.22 | 96.18 | 99.80 | 95.44 | 97.44 |
| **SMAFace** | 99.82 | 98.97 | 93.85 | 98.35 | 95.95 | 99.76 | 95.86 | 97.51 |
| **NSMAFace*** | 99.82 | 99.06 | 93.80 | 98.35 | 96.18 | 99.83 | 95.62 | 97.52 |
| MagFace | 99.83 | 98.46 | 92.87 | 98.17 | 96.15 | | | |
| PFC-0.3 | 99.83 | 99.23 | | 98.01 | | | | |
| AdaFace (Reproduce) | 99.83 | 99.04 | 94.45 | 97.88 | 96.05 | 99.80 | 95.94 | 97.57 |
| **SMAFace** | 99.83 | 99.11 | 94.70 | 97.93 | 96.10 | 99.76 | 95.94 | 97.63 |
| **SMAFace*** | 99.85 | 99.17 | 94.50 | 98.00 | 96.13 | 99.80 | 95.84 | 97.61 |
| **NSMAFace*** | 99.80 | 99.17 | 94.45 | 97.95 | 96.05 | 99.76 | 96.10 | 97.61 |

achieved SoTA performance on the CPLFW and VGG2-FP datasets, with accuracies of $94.70\%$ and $96.10\%$, respectively, surpassing all previous methods on these two datasets trained using all training datasets. The experimental results for IJB-B, IJB-C and TinyFace are presented in Appendix J, where they also demonstrate equal significance. The time complexity of our proposed method hasn't increased, with computational overhead not exceeding $1\%$. This is because our method didn't introduce new network architectures.

## 5 CONCLUSION

In this paper, we analyzed the mining-based method. By defining a fully controllable boundary and increasing the focus on hard and easy samples in the training dataset, we enhanced the performance of FR algorithms. The hyperparameters we designed are intuitive and fully controllable, and we visualized their effects. Our method offers a fresh perspective and provides a new direction for improving FR algorithms. Our experimental results effectively support our research content.

**Limitations** Despite the promising efficacy of the mining-based method, our research in this direction remains preliminary. We haven't made special accommodations for mislabeled samples, which may adversely impact training outcomes. The ideas presented in this paper might apply to broader image classification domains, though we haven't explored this potential. Incorporating it into domains beyond FR holds the potential for encouraging outcomes.

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

APPENDIX

## A   ADAPTIVE MARGIN FUNCTION

To further enhance the distinguishability between images, we adopt a margin function based on feature norm. Through this approach, we can enlarge the inter-class distance and reduce the intra-class distance, further boosting the model's discriminative power over image features. As we highlighted in Section 3.1, using an indicator function can balance the weights of easy and hard samples.

As a feature norm, $\|z_i\|$ can't be directly used as an image quality proxy; we normalize it with batch statistics $\mu_z$ and $\sigma_z$. Specifically, we set

$$\widehat{\|z_i\|} = \left\lfloor \frac{\|z_i\| - \mu_z}{\sigma_z/h} \right\rceil_{-1}^{1},$$

where $\mu_z$ and $\sigma_z$ represent the mean and standard deviation of all $\|z_i\|$ within a batch, respectively. $\lfloor \cdot \rceil$ clips the value between $-1$ and $1$, preventing gradient flow. This method ensures the distribution of $\widehat{\|z_i\|}$ approximates a unit Gaussian distribution as closely as possible, clipping its value between $-1$ and $1$ for data processing. The hyperparameter $h$ is used to control the concentration of the distribution and more precisely regulate the Gaussian distribution ratio between $-1$ and $1$. For its value, we directly adopt the optimal data after the ablation experiments of AdaFace(Kim et al., 2022), i.e., $h = 0.33$, ensure that the value of $\frac{\|z_i\| - \mu_z}{\sigma_z/h}$ falls between $-1$ and $1$ to the greatest extent.

It should be noted that, with smaller batch size, the stability of batch statistics, namely $\mu_z$ and $\sigma_z$, might be compromised, leading to potentially significant noise in the statistical data. Given that $\mu_z$ and $\sigma_z$ vary throughout the training phases, we adopt the same solution as AdaFace(Kim et al., 2022), which is the application of Exponential Moving Average (EMA) for $\mu_z$ and $\sigma_z$ to enhance stability. Let $\mu^{(k)}$ and $\sigma^{(k)}$ represent the batch statistics of $\|z_i\|$ at the $k^{th}$ step. We then have

$$\mu_z = \alpha\mu_z^{(k)} + (1 - \alpha)\mu_z^{(k-1)},$$

where $\alpha$ is the momentum coefficient and is set to $0.99$. $\sigma_z$ is computed in a similar manner.

In designing the margin function, we aim to increase the emphasis on samples as the quality of the image improves or decreases. To this end, we employ the functions $m_{angle}$ and $m_{add}$, representing the angular margin and the cosine margin respectively. Specifically, we define

$$f(\theta_j; m)_{\text{SM-CosFace}} = \begin{cases} s(\cos\theta_j - m_{\text{add}}) & j = y_i \\ s\cos\theta_j & j \neq y_i \end{cases}, \tag{10}$$

$$f(\theta_j; m)_{\text{SMAFace}} = \begin{cases} s(\cos(\theta_j + m_{\text{angle}}) - m_{\text{add}}) & j = y_i \\ s\cos\theta_j & j \neq y_i \end{cases}. \tag{11}$$

The $f(\theta_j; m)$ of SMAFace is consistent with AdaFace(Kim et al., 2022), integrating the mining-based method of AdaFace. We define $m_{\text{add}}$ and $m_{\text{angle}}$ as

$$m_{\text{angle}} = -m \cdot \widehat{\|z_i\|}, \quad m_{\text{add}} = m \cdot \widehat{\|z_i\|} + m.$$

It is worth noting that when $\widehat{\|z_i\|} = -1$, $f(\theta_j; m)_{\text{SM-CosFace}}$ becomes a function without margin, while $f(\theta_j; m)_{\text{SMAFace}}$ becomes akin to an ArcFace. When $\widehat{\|z_i\|} = 0$, both of them are reduced to the standard CosFace. For $\widehat{\|z_i\|} = 1$, $f(\theta_j; m)_{\text{SM-CosFace}}$ transforms into an enhanced version of CosFace, whereas $f(\theta_j; m)_{\text{SMAFace}}$ becomes a superimposition of negative-margin ArcFace and the enhanced CosFace.

## B   MARGIN-BASED SOFTMAX AND THE GRADIENT

We scrutinize the influence of gradient variations in traditional margin-based cosine similarity classification on the entire system. The mathematical form of the gradient of $\mathcal{L}_{\text{Margin}}$ with respect to

$\boldsymbol{W}_{:,j}$ and $\boldsymbol{x}_i$ is expressed as

$$\frac{\partial \mathcal{L}_{\text{Margin}}}{\partial \boldsymbol{W}_{:,j}} = \frac{\partial \mathcal{L}_{\text{Margin}}}{\partial p_j^{(i)}} \frac{\partial p_j^{(i)}}{\partial f(\cos\theta_j)} \frac{\partial f(\cos\theta_j)}{\partial \cos\theta_j} \frac{\partial \cos\theta_j}{\partial \boldsymbol{W}_{:,j}} = \left(p_j^{(i)} - \mathbf{1}_{\text{y}_i=j}\right) \frac{\partial f(\cos\theta_j)}{\partial \cos\theta_j} \frac{\left(\boldsymbol{x}_i^0 - \cos\theta_j \boldsymbol{W}_{:,j}^0\right)}{\|\boldsymbol{W}_{:,j}\|}, \tag{12}$$

$$\frac{\partial \mathcal{L}_{\text{Margin}}}{\partial \boldsymbol{x}_i} = \sum_{k=1}^{C} \frac{\partial \mathcal{L}_{\text{Margin}}}{\partial p_k^{(i)}} \frac{\partial p_k^{(i)}}{\partial f(\cos\theta_k)} \frac{\partial f(\cos\theta_k)}{\partial \cos\theta_k} \frac{\partial \cos\theta_k}{\partial \boldsymbol{x}_i} = \sum_{k=1}^{C} \left(p_k^{(i)} - \mathbf{1}_{\text{y}_i=k}\right) \frac{\partial f(\cos\theta_k)}{\partial \cos\theta_k} \frac{\left(\boldsymbol{W}_{:,k}^0 - \cos\theta_k \boldsymbol{x}_i^0\right)}{\|\boldsymbol{x}_i\|}.$$

For the proofs of Equations 12 and 13, refer to Appendix B.1 and B.2. From expressions Equations 12, we extract the scalar $\frac{\partial \mathcal{L}}{\partial p_j^{(i)}} \frac{\partial p_j^{(i)}}{\partial f(\cos\theta_j)} \frac{\partial f(\cos\theta_j)}{\partial \cos\theta_j}$, referred to as the Gradient Scale Parameter (GST) by Kim et al. (2022), denoted by the symbol $g$. For the GST based solely on the margin, its value is

$$g_{\text{Margin}} = \frac{\partial \mathcal{L}_{\text{Margin}}}{\partial p_j^{(i)}} \frac{\partial p_j^{(i)}}{\partial f(\cos\theta_j)} \frac{\partial f(\cos\theta_j)}{\partial \cos\theta_j} = \left(p_j^{(i)} - \mathbf{1}_{\text{y}_i=j}\right) \frac{\partial f(\cos\theta_j)}{\partial \cos\theta_j}. \tag{13}$$

GST fails to convey the appropriate degree of significance. The specific rationale behind this will become clearer as we delve into the analysis of vector directions later on. For the time being, let us temporarily set aside this matter for further discussion. It's noteworthy that this paper doesn't make improvements from the perspective of $j \neq y_i$. As such, our primary focus is on the $j = y_i$ case, analyzing the corresponding gradient scaling parameter $g$. We abbreviate $p_{y_i}^{(i)}$ as $p_{y_i}$ because the superscript $i$ is already reflected in the subscript $y_i$. Under normal circumstances, the expression for the normalized gradient scaling parameter $g_{\text{softmax}}$ is given by $g_{\text{softmax}} = (p_{y_i}-1)s$. It's easy to derive that the calculation result of $g_{\text{CosFace}}$ is identical to the above formula $g_{\text{CosFace}} = (p_{y_i}-1)s$. However, we found that the value of $g_{\text{ArcFace}}$ varies with the difficulty of the samples. As $\theta_{y_i}$ increases, $g_{\text{ArcFace}}$ shows a decreasing trend

$$g_{\text{ArcFace}} = (p_{y_i} - 1)s\left(\cos(m) + \frac{\cos\theta_{y_i}\sin(m)}{\sqrt{1-\cos^2\theta_{y_i}}}\right). \tag{14}$$

This outcome aligns with our expectations, and we will delve into a more detailed analysis in subsequent sections. For the proof of Equation 14, refer to Appendix B.3.

## B.1   CALCULATION OF GST

To calculate $g_{\text{Margin}}$, the first step is to determine the value of $\frac{\partial \mathcal{L}_{\text{Margin}}}{\partial p_j^{(i)}}$. According to the definition of the margin-based loss function $\mathcal{L}_{\text{Margin}}$, its expression can be written as

$$\mathcal{L}_{\text{Margin}} = -\log(p_j^{(i)}), \tag{15}$$

from which we can derive that

$$\frac{\partial \mathcal{L}_{\text{Margin}}}{\partial p_j^{(i)}} = -\frac{1}{p_j^{(i)}}.$$

Next, we revisit the definition of $p_j^{(i)}$, which represents the output probability of class $j$ after input $x_i$ has been processed by the softmax function. Its expression can be written as

$$p_j^{(i)} = \frac{\exp(f(\theta_{y_i}; m))}{\exp(f(\theta_{y_i}; m)) + \sum_{j \neq y_i}^{N} \exp(s\cos\theta_j)}, \tag{16}$$

where $f(\theta_{y_i}; m) = f(\cos\theta_j)$, and their difference is merely notational. Subsequently, we want to solve $\frac{\partial p_j^{(i)}}{\partial f(\cos\theta_j)}$, which involves handling two cases.

For the case where $j = y_i$, using the quotient rule for derivatives, we can let $u = \exp(f(\theta_{y_i}; m))$, $v = \exp(f(\theta_{y_i}; m)) + \sum_{j \neq y_i}^{N} \exp(s\cos\theta_j)$, and then obtain

$$u' = \frac{\partial u}{\partial f(\cos\theta_{y_i})} = \exp(f(\cos\theta_{y_i})) = p_{y_i}v, v' = \frac{\partial v}{\partial f(\cos\theta_{y_i})} = \exp(f(\cos\theta_{y_i})) = p_{y_i}v.$$

Furthermore, $u$ can be expressed as

$$u = p_{y_i} v.$$

Based on the analysis above, we can calculate the specific value of $\frac{\partial p_{y_i}}{\partial f(\cos \theta_{y_i})}$ as

$$\frac{\partial p_{y_i}}{\partial f(\cos \theta_{y_i})} = \frac{u'v - uv'}{v^2} = \frac{p_{y_i} v^2 - p_{y_i} v \cdot p_{y_i} v}{v^2} = p_{y_i} \left(1 - p_{y_i}\right). \tag{17}$$

The other case is where $j \neq y_i$. In this situation, we know that $f(\theta_j; m)$ is a piecewise function. When $j \neq y_i$, its value satisfies $f(\theta_j; m) = s \cos \theta_j$, hence we get

$$u' = \frac{\partial u}{\partial f(\cos \theta_j)} = \frac{\partial u}{\partial (s \cos \theta_j)} = 0,$$

$$v' = \frac{\partial v}{\partial f(\cos \theta_j)} = \frac{\partial v}{\partial (s \cos \theta_j)} = \exp(f(s \cos \theta_j)) = \exp(f(\cos \theta_j)) = p_j^{(i)} v.$$

Following the previous method, we can derive

$$\frac{\partial p_j^{(i)}}{\partial f(\cos \theta_j)} = \frac{u'v - uv'}{v^2} = \frac{0 - p_j^{(i)} v \cdot p_j^{(i)} v}{v^2} = p_j^{(i)} \left(0 - p_j^{(i)}\right). \tag{18}$$

Summarizing the analysis above, we find that, regardless of whether the value of $j$ is $y_i$ or not, the value of $\frac{\partial p_j^{(i)}}{\partial f(\cos \theta_j)}$ can be expressed as

$$\frac{\partial p_j^{(i)}}{\partial f(\cos \theta_j)} = \left(\mathbf{1}_{y_i=j} - p_j^{(i)}\right) p_j^{(i)}. \tag{19}$$

Finally, we also need to consider the value of $\frac{\partial f(\cos \theta_j)}{\partial \cos \theta_j}$, which involves the specific function $f(\cos \theta_j)$. Notably, for softmax and CosFace, we have the following equation

$$\frac{\partial f(\cos \theta_j)}{\partial \cos \theta_j} = s.$$

This result can be directly observed without detailed derivation. As for the ArcFace case, we have given the proof process for $j = y_i$ in the previous section, and the derivation process for $j \neq y_i$ is identical. Therefore, we can obtain

$$\begin{aligned} g_{\text{Margin}} &= \frac{\partial \mathcal{L}_{\text{Margin}}}{\partial p_j^{(i)}} \frac{\partial p_j^{(i)}}{\partial f(\cos \theta_j)} \frac{\partial f(\cos \theta_j)}{\partial \cos \theta_j} = -\frac{1}{p_j^{(i)}} \left(\mathbf{1}_{y_i=j} - p_j^{(i)}\right) p_j^{(i)} \frac{\partial f(\cos \theta_j)}{\partial \cos \theta_j} \\ &= \left(p_j^{(i)} - \mathbf{1}_{y_i=j}\right) \frac{\partial f(\cos \theta_j)}{\partial \cos \theta_j}. \end{aligned} \tag{20}$$

## B.2 COMPARISON OF THE DERIVATIVES OF $\mathcal{L}_{\text{MARGIN}}$

We begin by revisiting the derivatives of the margin loss function $\mathcal{L}_{\text{Margin}}$ concerning $\boldsymbol{W}_{:,j}$ and $\boldsymbol{x}_i$, expressions for which have already been detailed in the paper

$$\frac{\partial \mathcal{L}_{\text{Margin}}}{\partial \boldsymbol{W}_{:,j}} = \left(p_j^{(i)} - \mathbf{1}_{y_i=j}\right) \frac{\partial f(\cos \theta_j)}{\partial \cos \theta_j} \frac{\partial \cos \theta_j}{\partial \boldsymbol{W}_{:,j}}, \quad \frac{\partial \mathcal{L}_{\text{Margin}}}{\partial \boldsymbol{x}_i} = \sum_{k=1}^{C} \left(p_k^{(i)} - \mathbf{1}_{y_i=k}\right) \frac{\partial f(\cos \theta_k)}{\partial \cos \theta_k} \frac{\partial \cos \theta_k}{\partial \boldsymbol{x}_i}. \tag{21}$$

Examining these two expressions, we note that the computation of $\frac{\partial \mathcal{L}_{\text{Margin}}}{\partial \boldsymbol{W}_{:,j}}$ does not involve any summation, while $\frac{\partial \mathcal{L}_{\text{Margin}}}{\partial \boldsymbol{x}_i}$ requires summation over all classes $k$. This discrepancy can be readily understood.

First, consider $\frac{\partial \mathcal{L}_{\text{Margin}}}{\partial \boldsymbol{W}_{:,j}}$, the gradient of the loss function $\mathcal{L}_{\text{Margin}}$ with respect to the $j^{\text{th}}$ column of the weight matrix $\boldsymbol{W}$. In the forward propagation, the $j^{\text{th}}$ column of $\boldsymbol{W}$ interacts solely with the

corresponding class $j$ logic, focusing mainly on the specific computation path for class $j$. Since $\boldsymbol{W}_{:,j}$ does not directly influence the output for non-$j$ classes, during backpropagation, only those gradient parts directly related to $\boldsymbol{W}_{:,j}$ need to be considered, without summation across all classes.

However, when we consider the gradient of the loss function concerning the input sample $\boldsymbol{x}_i$, denoted by $\frac{\partial \mathcal{L}_{\text{Margin}}}{\partial \boldsymbol{x}_i}$, the scenario is different. During forward propagation, the input sample $\boldsymbol{x}_i$ is directly involved in the probability computation for all classes, with each similarity score impacting the final loss function. Therefore, the gradient must be summed across all classes to ensure we capture the complete information on how changes in $\boldsymbol{x}_i$ affect the loss. This summation can be seen as an aggregation step, encompassing the influence of all classes on the input sample $\boldsymbol{x}_i$. This step is crucial in backpropagation, providing a comprehensive view of how to adjust $\boldsymbol{x}_i$ to minimize the loss.

Next, we focus on the final term of the expression, namely $\frac{\partial \cos \theta_j}{\partial \boldsymbol{W}_{:,j}}$. It is well-known that the cosine similarity $\cos \theta_j$ is given by the dot product of vectors $\boldsymbol{x}_i$ and $\boldsymbol{W}_{:,j}$ divided by the magnitudes of the two vectors, expressed as

$$\cos \theta_j = \frac{\boldsymbol{x}_i \cdot \boldsymbol{W}_{:,j}}{\|\boldsymbol{x}_i\| \|\boldsymbol{W}_{:,j}\|}. \tag{22}$$

Our task is to compute the derivative of this ratio concerning the vector $\boldsymbol{W}_{:,j}$. Since both the numerator and denominator are functions of $\boldsymbol{W}_{:,j}$, we must apply the quotient rule for differentiation. For ease of manipulation, we rewrite $\cos \theta_j$ as the ratio of two functions, $p(\boldsymbol{W}_{:,j}) = \boldsymbol{x}_i \cdot \boldsymbol{W}_{:,j}$ and $q(\boldsymbol{W}_{:,j}) = \|\boldsymbol{x}_i\| \|\boldsymbol{W}_{:,j}\|$. Hence, $\cos \theta_j = \frac{p(\boldsymbol{W}_{:,j})}{q(\boldsymbol{W}_{:,j})}$. Subsequently, the derivative is found to be

$$\frac{\partial \cos \theta_j}{\partial \boldsymbol{W}_{:,j}} = \frac{\frac{\partial p(\boldsymbol{W}_{:,j})}{\partial \boldsymbol{W}_{:,j}} q(\boldsymbol{W}_{:,j}) - p(\boldsymbol{W}_{:,j}) \frac{\partial q(\boldsymbol{W}_{:,j})}{\partial \boldsymbol{W}_{:,j}}}{q(\boldsymbol{W}_{:,j})^2}.$$

In the subsequent step, the task is to compute $\frac{\partial p(\boldsymbol{W}_{:,j})}{\partial \boldsymbol{W}_{:,j}}$ and $\frac{\partial q(\boldsymbol{W}_{:,j})}{\partial \boldsymbol{W}_{:,j}}$. Initially, $\frac{\partial p(\boldsymbol{W}_{:,j})}{\partial \boldsymbol{W}_{:,j}}$ equals $\boldsymbol{x}_i$ since the derivative of $\boldsymbol{x}_i \cdot \boldsymbol{W}_{:,j}$ with respect to $\boldsymbol{W}_{:,j}$ is $\boldsymbol{x}_i$. Secondly, for $q(\boldsymbol{W}_{:,j}) = \|\boldsymbol{x}_i\| \|\boldsymbol{W}_{:,j}\|$, we solely consider the derivative of $\|\boldsymbol{W}_{:,j}\|$ with respect to $\boldsymbol{W}_{:,j}$. The derivative of a vector's magnitude concerning the vector itself is the vector divided by its magnitude, i.e., $\frac{\boldsymbol{W}_{:,j}}{\|\boldsymbol{W}_{:,j}\|}$. Consequently, $\frac{\partial q(\boldsymbol{W}_{:,j})}{\partial \boldsymbol{W}_{:,j}} = \frac{\boldsymbol{W}_{:,j}}{\|\boldsymbol{W}_{:,j}\|} \|\boldsymbol{x}_i\|$. Substituting these results into the quotient rule for differentiation, we acquire

$$\frac{\partial \cos \theta_j}{\partial \boldsymbol{W}_{:,j}} = \frac{\|\boldsymbol{x}_i\| \|\boldsymbol{W}_{:,j}\| \boldsymbol{x}_i - \boldsymbol{x}_i \cdot \boldsymbol{W}_{:,j} \frac{\boldsymbol{W}_{:,j}}{\|\boldsymbol{W}_{:,j}\|} \|\boldsymbol{x}_i\|}{(\|\boldsymbol{x}_i\| \|\boldsymbol{W}_{:,j}\|)^2} = \frac{\boldsymbol{x}_i}{\|\boldsymbol{x}_i\| \|\boldsymbol{W}_{:,j}\|} - \frac{\cos \theta_j \boldsymbol{W}_{:,j}}{\|\boldsymbol{W}_{:,j}\|^2}.$$

Here, the unit vector $\frac{\boldsymbol{x}_i}{\|\boldsymbol{x}_i\|}$ is denoted as $\boldsymbol{x}_i^0$, and the unit vector $\frac{\boldsymbol{W}_{:,j}}{\|\boldsymbol{W}_{:,j}\|}$ is denoted as $\boldsymbol{W}_{:,j}^0$. The aforementioned equation can be represented as

$$\frac{\partial \cos \theta_j}{\partial \boldsymbol{W}_{:,j}} = \frac{\left(\boldsymbol{x}_i^0 - \cos \theta_j \boldsymbol{W}_{:,j}^0\right)}{\|\boldsymbol{W}_{:,j}\|}. \tag{23}$$

From this equation, it can be discerned that the numerator represents the unit vector in the direction of $\boldsymbol{x}_i$ subtracted by its projection in the direction of $\boldsymbol{W}_{:,j}$, culminating in a vector that is perpendicular to $\boldsymbol{W}_{:,j}$. The magnitude of this vector equals the distance from the endpoint of $\boldsymbol{x}_i^0$ to $\boldsymbol{W}_{:,j}$, i.e.

$$\left\|\frac{\partial \cos \theta_j}{\partial \boldsymbol{W}_{:,j}}\right\| = \frac{\sin \theta_j}{\|\boldsymbol{W}_{:,j}\|}. \tag{24}$$

When the directions of these two vectors are identical, the numerator will be a zero vector. Similarly, the derivative with respect to $\boldsymbol{x}_i$ can be deduced as

$$\frac{\partial \cos \theta_k}{\partial \boldsymbol{x}_i} = \frac{\left(\boldsymbol{W}_{:,k}^0 - \cos \theta_k \boldsymbol{x}_i^0\right)}{\|\boldsymbol{x}_i\|}, \left\|\frac{\partial \cos \theta_k}{\partial \boldsymbol{x}_i}\right\| = \frac{\sin \theta_k}{\|\boldsymbol{x}_i\|}. \tag{25}$$

As can be seen from the direction of $\left(\boldsymbol{W}_{:,k}^0 - \cos \theta_k \boldsymbol{x}_i^0\right)$ and $\left(\boldsymbol{x}_i^0 - \cos \theta_j \boldsymbol{W}_{:,j}^0\right)$, it has a positive effect only when the value of $\left(p_j^{(i)} - \mathbf{1}_{\text{y}_i=\text{j}}\right) \frac{\partial f(\cos \theta_j)}{\partial \cos \theta_j}$ and $\left(p_k^{(i)} - \mathbf{1}_{\text{y}_i=\text{k}}\right) \frac{\partial f(\cos \theta_k)}{\partial \cos \theta_k}$ is negative, and

the $st$ we provide in our text perfectly meets this requirement. The $st$ has the opposite sign to $\left(p_j^{(i)} - \mathbf{1}_{y_i=j}\right)\frac{\partial f(\cos\theta_j)}{\partial\cos\theta_j}$ and $\left(p_k^{(i)} - \mathbf{1}_{y_i=k}\right)\frac{\partial f(\cos\theta_k)}{\partial\cos\theta_k}$, so it can be used to represent the level of importance for the samples, while GST cannot meet this requirement. In some cases, such as when the samples are of low quality and hard, $st$ may be negative, which means that we are giving up on these worthless samples.

### B.3 Derivation of Angular Margin

The margin function expression of ArcFace is given as

$$f(\cos\theta_{y_i}) = s\cos(m + \theta_{y_i}), \tag{26}$$

where $s$ represents the scale factor, $\theta_{y_i}$ denotes the angle, and $m$ is the increment of the cosine margin. To calculate the derivative concerning $\cos\theta_{y_i}$, we need to apply the chain rule. Moreover, based on $\frac{d\cos x}{dx} = -\sin x$, we can derive

$$\frac{dx}{d\cos x} = -\frac{1}{\sqrt{1-\cos^2 x}}.$$

Consequently, we can deduce

$$\frac{\partial f(\cos\theta_{y_i})}{\partial\cos\theta_{y_i}} = \frac{\partial f(\cos\theta_{y_i})}{\partial\theta_{y_i}}\frac{\partial\theta_{y_i}}{\partial\cos\theta_{y_i}} = -s\sin(m + \arccos(\cos\theta_{y_i}))\frac{1}{\sqrt{1-\cos^2\theta_{y_i}}}.$$

Next, utilizing the trigonometric identity $\sin(a+b) = \sin a\cos b + \cos a\sin b$, we can decompose it as follows

$$\frac{\partial f(\cos\theta_{y_i})}{\partial\cos\theta_{y_i}} = -s(\sin(m)\cos(\arccos(\cos\theta_{y_i})) + \cos(m)\sin(\arccos(\cos\theta_{y_i})))\frac{1}{\sqrt{1-\cos^2\theta_{y_i}}}.$$

With the basic inverse trigonometric identities $\cos(\arccos(x)) = x$ and $\sin(\arccos(x)) = \sqrt{1-x^2}$, we obtain

$$\frac{\partial f(\cos\theta_{y_i})}{\partial\cos\theta_{y_i}} = -s\left(\cos(m) + \frac{\cos\theta_{y_i}\sin(m)}{\sqrt{1-\cos^2\theta_{y_i}}}\right). \tag{27}$$

Lastly, substituting the above result into the equation $g_{\text{ArcFace}} = (p_{y_i} - 1)\frac{\partial f(\cos\theta_{y_i})}{\partial\cos\theta_{y_i}}$ yields

$$g_{\text{ArcFace}} = (p_{y_i} - 1)s\left(\cos(m) + \frac{\cos\theta_{y_i}\sin(m)}{\sqrt{1-\cos^2\theta_{y_i}}}\right). \tag{28}$$

With this, the whole derivation process is complete. QED.

## C GST of SM-CosFace

We adopt the following step-by-step calculation for $g_{\text{SM-CosFace}}$

$$\frac{\partial\mathcal{L}_{\text{Mining}}}{\partial p_j^{(i)}} = -\left(\frac{\partial I(p_j^{(i)})}{\partial p_j^{(i)}}\log p_j^{(i)} + \frac{I(p_j^{(i)})}{p_j^{(i)}}\right). \tag{29}$$

Additionally, the value of $\frac{\partial p_j^{(i)}}{\partial f(\cos\theta_j)}$ remains the same as that with $\mathcal{L}_{\text{Margin}}$, as shown below

$$\frac{\partial p_j^{(i)}}{\partial f(\cos\theta_j)} = \left(\mathbf{1}_{y_i=j} - p_j^{(i)}\right)p_j^{(i)}. \tag{30}$$

At this step, we observe that $\widehat{\|z_i\|}$ is a scalar with respect to $p_j^{(i)}$, which implies

$$\frac{\partial f(\cos\theta_j)}{\partial\cos\theta_j} = s. \tag{31}$$

Summarizing the above steps, for the case where $j = y_i$, we can derive the gradient scaling factor $g_{\text{SM-CosFace}}$ as follows

$$
\begin{aligned}
g_{\text{SM-CosFace}} &= -\left( \frac{\partial I(p_{y_i})}{\partial p_{y_i}} \log p_{y_i} + \frac{I(p_{y_i})}{p_{y_i}} \right) (1 - p_{y_i}) p_{y_i} s \\
&= \left( \frac{\partial I(p_{y_i})}{\partial p_{y_i}} p_{y_i} \log p_{y_i} + I(p_{y_i}) \right) (p_{y_i} - 1) s,
\end{aligned}
\tag{32}
$$

## D  DERIVATION OF $I(p_{y_i})$

In order to conveniently calculate $\frac{\partial I(p_{y_i})}{\partial p_{y_i}}$, we introduce a new expression for the indicator function $I(p_{y_i})$, where the $sigmoid$ function is defined as follows

$$
\sigma(x) = \frac{1}{1 + e^{-x}}.
$$

Hence, we can express $I(p_{y_i})$ as

$$
I(p_{y_i}) = 1 + \alpha \left( \sigma(p_0 - p_{y_i}) - 0.5 \right).
\tag{33}
$$

Considering the following commonly used result

$$
\frac{\partial \sigma(x)}{\partial x} = \sigma(x)(1 - \sigma(x)),
$$

we can derive $\frac{\partial I(p_{y_i})}{\partial p_{y_i}}$ as

$$
\frac{\partial I(p_{y_i})}{\partial p_{y_i}} = \alpha \sigma(p_0 - p_{y_i})(\sigma(p_0 - p_{y_i}) - 1).
\tag{34}
$$

## E  ANALYSIS OF SM-COSFACE AND ARCFACE

We have plotted the graph concerning $p_{y_i}$, illustrating the changes in $st_{\text{SM-CosFace}}$ as shown in Figure 3. By observing the curve in the figure, it can be discerned that different $st$ values emerge under varying difficulties, which aligns with our expectations. Additionally, we have separately observed the effects of parameters $p_0$ and $\alpha$ on $st$. The results reveal that a larger $\alpha$ implies a more pronounced disparity in the attention given to samples under easy and hard scenarios. Meanwhile, alterations in $p_0$ will influence the overall penalty magnitude.

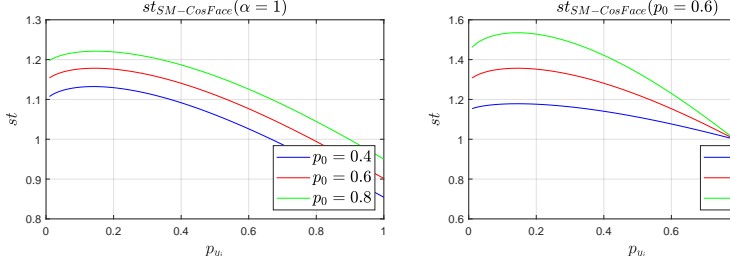

Figure 3: This is the plot of the scaling term function of SM-CosFace. When the function plot parameters of $st_{\text{SM-CosFace}}$ regarding $p_{y_i}$ and $\alpha$ are set to $\alpha = 1$, by changing the magnitude of $p_0$, it's not difficult to see that the penalty for all samples has been increased. Meanwhile, when $p_0 = 0.6$, changing the magnitude of $\alpha$ can further enhance the gap between easy samples and hard samples. Both the $\widehat{\|z_i\|}$ are set to 0.

We further plotted the graph concerning $\cos \theta_{y_i}$, comparing the variations between $st_{\text{ArcFace}}$ and $st_{\text{SM-CosFace}}$, as illustrated in Figure 4. It is evident from the figure that ArcFace imposes relatively

extreme penalties or rewards for samples at very high difficulties (where $\cos \theta_{y_i}$ approaches $-1$) and very low difficulties (where $\cos \theta_{y_i}$ nears 1). In contrast, SM-CosFace more gently scales the hard and easy samples. Furthermore, our method boasts fully controllable boundaries. Specifically, by adjusting the value of $\alpha$, we can effectively modulate its operational range, especially in exceptionally difficult or simple scenarios. We have also displayed the surface of $st$ concerning the margin $m$ and $\cos \theta_{y_i}$, as depicted in Figure 5. It can be observed that the positive and negative values of $m$ have opposite effects for ArcFace, whereas for SM-CosFace, it only alters the boundaries, marking a distinction between the two.

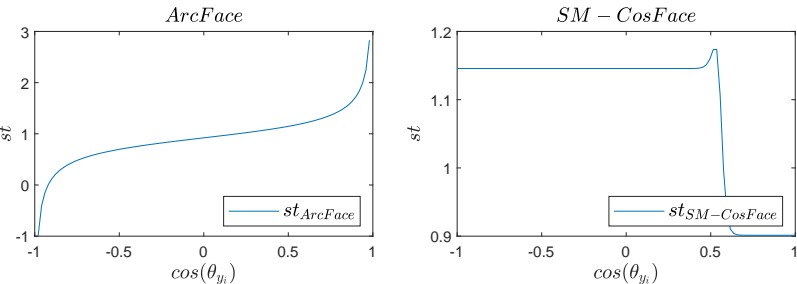

Figure 4: This is the plot of the scaling term function of ArcFace and SM-CosFace. The plot of $st_{\text{ArcFace}}$ with respect to $\cos \theta_{y_i}$ has a parameter $m = 0.4$. The plot of $st_{\text{SM-CosFace}}$ with respect to $\cos \theta_{y_i}$ has parameters set as $p_0 = 0.6$, $\alpha = 1.0$, $m = 0.4$ and $s = 64.0$. We set $\widehat{\|z_i\|}$ to 0. In actual training, we do use $s = 64.0$, because we need to take into account factors such as the number of samples.

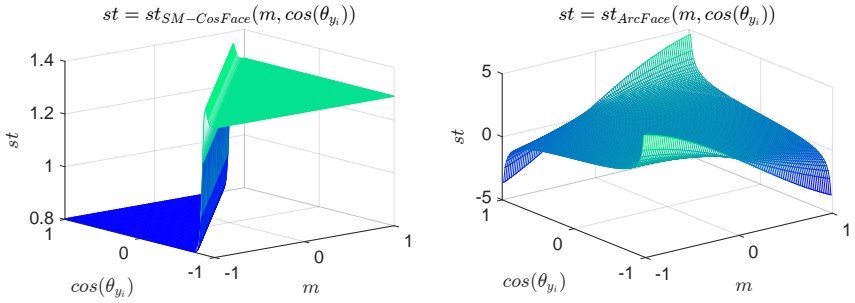

Figure 5: This is the plot of the scaling term function of ArcFace and SM-CosFace. The plot of $st_{\text{SM-CosFace}}$ with respect to $\cos \theta_{y_i}$ and $m$ has parameters set as $p_0 = 0.6$, $\alpha = 2.0$ and $s = 64.0$. $\widehat{\|z_i\|}$ remains 0. Both $m$ and $\cos \theta_{y_i}$ vary within the interval $[-1, 1]$.

Although SM-CosFace is more controllable than ArcFace, it still falls short in handling both easy and hard samples perfectly, which is why we opt for SMAFace.

## F  Scaling Term of SMAFace

In our paper, we have analyzed SM-CosFace. Accordingly, SMAFace can be derived by multiplying SM-CosFace with the $st$ of AdaFace. This allows us to easily understand its properties. In practical use, we often employ SMAFace*, which can be regarded as the normalized SMAFace. The indicator function of SMAFace* deviates from that of SMAFace by a factor of $\left\| \frac{7}{7+\alpha} \right\|$. We will start our analysis with NSMAFace* where $\alpha < 0$, and extend to the case where $\alpha > 0$.

For $\alpha = -1, \alpha = -2, \alpha = -4, \alpha = -8, \alpha = -32$, we have plotted the $st_{\text{NSMAFace}*}$ for each, as shown in Figure 6. It can be seen that when $\alpha$ is adjusted to $-16$, its trend becomes uncontrolled, and the leftmost $st$ even becomes positive, which is illogical and harmful to the performance of the

FR algorithm. When $\alpha = -32$, it can be interpreted as emphasizing both hard and easy samples, while neglecting average samples.

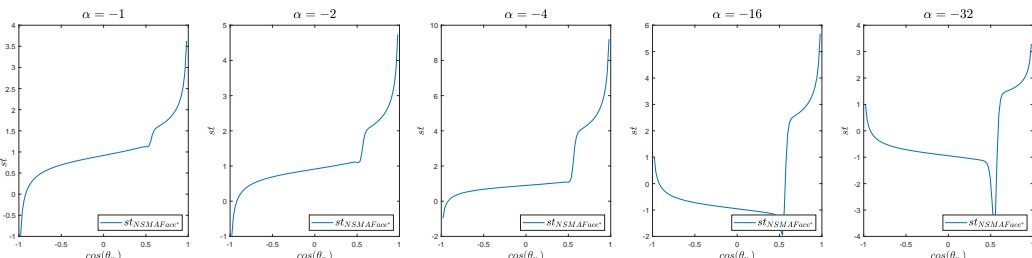

Figure 6: The $st_{\text{NSMAFace}^*}$ with $\widehat{\|z_i\|} = 0, m = 0.4, s = 64, p_0 = 0.6$ and $\alpha < 0$ being far from $-7$.

Having fully understood the above, we now consider a different situation when $\alpha > 0$, where the scenario becomes entirely different, resulting in a bimodal picture. As shown in Figure 7, it's not difficult to find that when $\alpha$ is too large, it is unreasonable as the rightmost $st_{\text{SMAFace}^*}$ is less than $0$, which is detrimental to the training results.

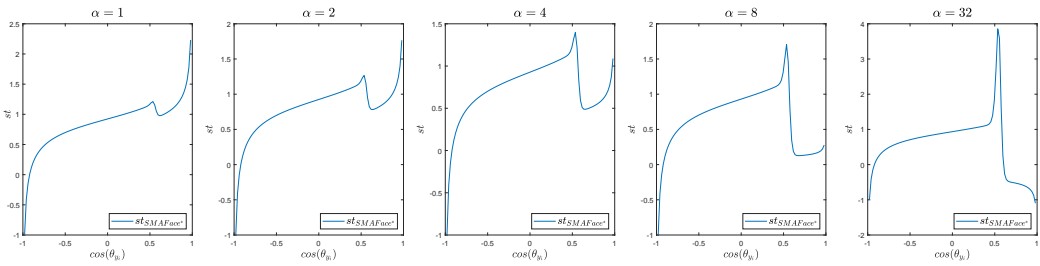

Figure 7: The $st_{\text{SMAFace}^*}$ with $\widehat{\|z_i\|} = 0, m = 0.4, s = 64, p_0 = 0.6$ and $\alpha > 0$.

For the situation where $\alpha$ is near $-7$, the value of NSMAFace$^*$ becomes too large and is not suitable for training, as shown in Figure 8. In this case, its training method is too dismissive of high-quality samples, making it difficult to train an effective FR model.

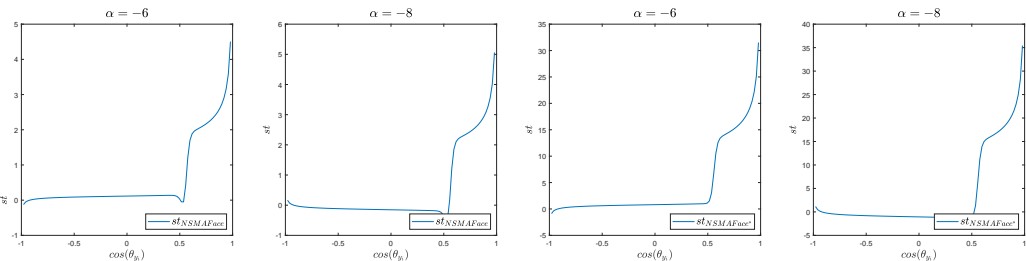

Figure 8: The $st_{\text{NSMAFace}^*}$ and $st_{\text{NSMAFace}}$ with $\widehat{\|z_i\|} = 0, m = 0.4, s = 64, p_0 = 0.6$ and $\alpha$ approaching $-7$.

## G  OTHER MINING-BASED METHOD

In our previous work, we mentioned that F-Softmax was proposed by Lin et al. (2017) and HM-Softmax by Shrivastava et al. (2016). Their specific forms are given by

$$I_{HM-Softmax}(p_{y_i}) = \begin{cases} 0 & p_{y_i} < 0.5 \\ 1 & p_{y_i} \geq 0.5 \end{cases}, I_{F-Softmax}(p_{y_i}) = (1 - p_{y_i})^{\gamma}. \tag{35}$$

The expression, $\frac{\partial \mathcal{L}_{\text{Mining}}}{\partial p_j^{(i)}} = -\left(\frac{\partial I(p_j^{(i)})}{\partial p_j^{(i)}} \log p_j^{(i)} + \frac{I(p_j^{(i)})}{p_j^{(i)}}\right)$, always hold. Thus, substituting $I(p_j^{(i)})$ with $I_{F-Softmax}(p_{y_i})$ and $I_{HM-Softmax}(p_{y_i})$ gives us their respective $st$ values. Let $f(\theta_{y_i}; m) = s \cos \theta_{y_i}$, we have

$$st_{\text{HM-Softmax}} = I_{HM-Softmax}(p_{y_i}) = \begin{cases} 0 & p_{y_i} < 0.5 \\ 1 & p_{y_i} \geq 0.5 \end{cases}, \tag{36}$$

$$\begin{aligned} st_{\text{F-Softmax}} &= \frac{\partial I_{F-Softmax}(p_{y_i})}{\partial p_{y_i}} p_{y_i} \log p_{y_i} + I_{F-Softmax}(p_{y_i}) \\ &= \gamma(1 - p_{y_i})^{\gamma-1} p_{y_i} \log p_{y_i} + (1 - p_{y_i})^{\gamma}. \end{aligned} \tag{37}$$

From the expressions alone, we cannot discern much. It is essential to consider the definition of $p_{y_i}$, given by

$$p_{y_i} = \frac{\exp(f(\theta_{y_i}; m))}{\exp(f(\theta_{y_i}; m)) + \sum_{j \neq y_i}^{N} \exp(s \cos \theta_j)}.$$

Combining the above equation, we can analyze their properties. The optimal value for $s$ was found to be $64$, and we use $s = 64$ directly during training unless the adaptive strategy proposed by Ada-Cos(Zhang et al., 2019) is adopted. Blindly using a value of $s$ that is too large results in insufficient penalization for misclassified samples, whereas a value that is too small penalizes samples that are already correctly classified. This is precisely the reason why we do not vary its value. Consequently, the relationship between $p_{y_i}$ and $\cos \theta_{y_i}$ appears as a piecewise function: it is $0$ when $\cos \theta_{y_i}$ is below a certain threshold and $1$ when above. In reality, this is an illusion. With $s = 64$, the transition of $p_{y_i}$ from $0$ to $1$ is extremely rapid, making it seem like a jump. However, this directly results in the scenario depicted in Figure 9, where the $st$ values for both HM-Softmax and F-Softmax are $0$ when $\cos \theta_{y_i}$ is above a certain boundary.

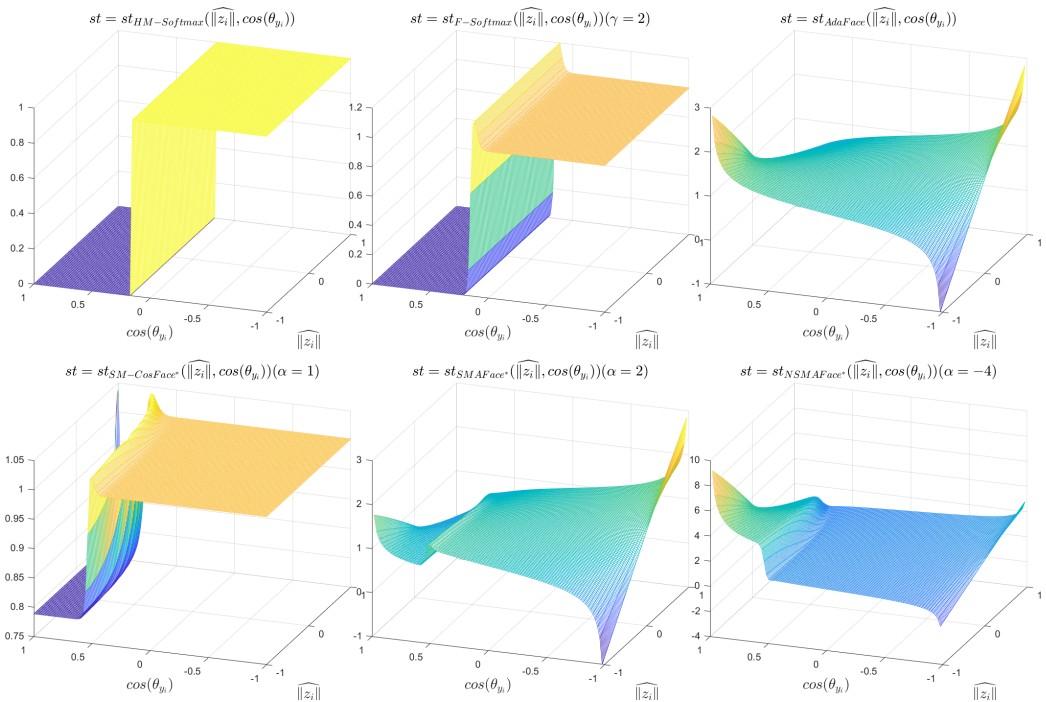

Figure 9: Comparison of HM-Softmax, F-Softmax, AdaFace, SM-CosFace, SMAFace* and NSMAFace* concerning $m_{\text{angle}}$.

## H  VARIANT OF SMAFACE

We have further adjusted the indicator function based on the original basis. We first review the expression of $I^*(p_{y_i})$

$$I^*(p_{y_i}) = \left\| \frac{7}{7+\alpha} \right\| \left( 1 + \alpha \left( \frac{1}{1+e^{p_{y_i}-p_0}} - 0.5 \right) \right). \tag{38}$$

For it, we make the following adjustments

$$I^*(p_{y_i})_{\text{weak}} = \left\| \frac{7}{7+\alpha} \right\| \left( 1 + \alpha \left( \frac{1}{1+e^{(p_{y_i}-p_0)^-}} - 0.5 \right) \right),$$

$$I^*(p_{y_i})_{\text{strong}} = \left\| \frac{7}{7+\alpha} \right\| \left( 1 + \alpha \left( \frac{1}{1+e^{(p_{y_i}-p_0)^+}} - 0.5 \right) \right). \tag{39}$$

Wherein, $(p_{y_i} - p_0)^-$ represents $\min(p_{y_i} - p_0, 0)$, and $(p_{y_i} - p_0)^+$ represents $\max(p_{y_i} - p_0, 0)$. Then, its properties change. We name them strong and weak not because of their higher or lower values, but based on whether the terrain features in the image are more prominent or flatter. We have shown the impact it brings, as shown in Figure 10. It can be seen that when $\alpha = 2$ if we use the normal $I^*(p_{y_i})$, its emphasis on easy samples is insufficient. However, if we use $I^*(p_{y_i})_{\text{strong}}$, the distinction for all samples is not enough. After using $I^*(p_{y_i})_{\text{weak}}$, we can have something akin to a bimodal emphasis. When $\alpha = 3, 4$, the situation is similar to that of $\alpha = 2$, just the trend is less gentle.

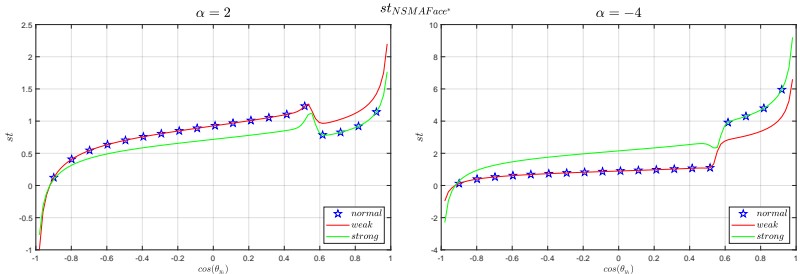

Figure 10: The $st$ graph of NSMAFace*, NSMAFace$^*_{\text{strong}}$ and NSMAFace$^*_{\text{weak}}$ with $\|\widehat{z_i}\| = 0, m = 0.4, s = 64, p_0 = 0.6$. The left $\alpha$ is 2 and the right $\alpha$ is $-4$.

In addition, there is another variant, for which we provide the indicator function

$$I^*(p_{y_i}, \beta) = \left\| \frac{7}{7+\alpha} \right\| \left( 1 + \alpha_0 \left( \frac{1}{1+e^{\beta(p_{y_i}-p_0)}} - 0.5 \right) \right), \alpha_0 = 1. \tag{40}$$

The role of $\beta$ is similar to that of $\alpha$, and its effect is within our expectations. In Figure 11 and 12, we can observe that it can also achieve parameter scaling and yield excellent results.

## I  TESTING SETS AND THRESHOLD

As shown in Table 3, datasets CALFW and VGG2-FP(Zheng et al., 2017; Cao et al., 2018) are more conducive for SMAFace and NSMAFace training strategies. In these testing sets, mining-based methods display inherent advantages. Even without incorporating AdaFace's margin function or, in other words, without angular margin, it still manages decent accuracy. This could be due to the pronounced difficulty variation in these face images or possibly because they are tough among the seven HQ Quality datasets, highlighting our method's distinct advantages.

Through a meticulous examination of the data presented in Table 4 within the supplementary materials, it has been ascertained that the optimal value for the parameter $p_0$ resides in the proximity of 0.6. This observation is underpinned by the recognition that samples with exceedingly minute probabilities are susceptible to potential misclassification, necessitating cautious consideration rather than undue emphasis. This empirical finding is in complete consonance with the overarching conclusions expounded within the main body of the paper.

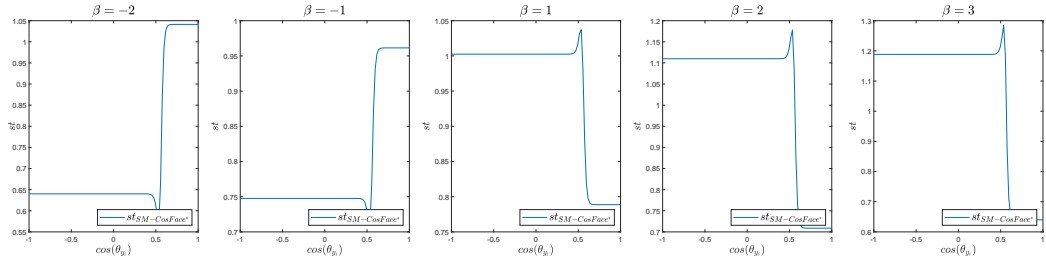

Figure 11: The $st$ graph of SM-CosFace$^*_{\text{weak}}$ and NSM-CosFace$^*_{\text{weak}}$ with $\widehat{\|z_i\|} = 0, m = 0.4, s = 64, p_0 = 0.6, \alpha_0 = 1$.

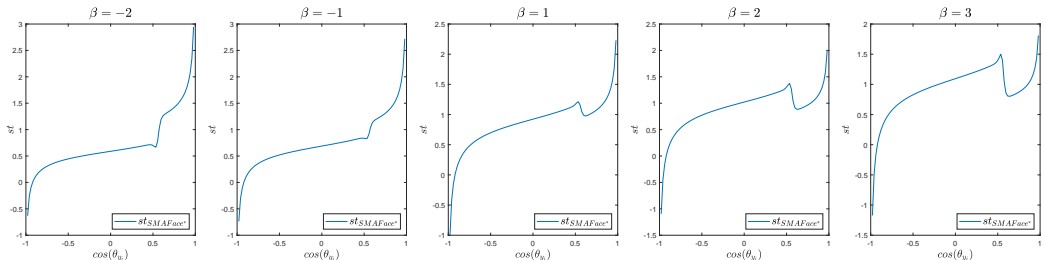

Figure 12: The $st$ graph of SMAFace$^*_{\text{weak}}$ and NSMAFace$^*_{\text{weak}}$ with $\widehat{\|z_i\|} = 0, m = 0.4, s = 64, p_0 = 0.6, \alpha_0 = 1$.

## J  COMPARISON WITH SoTA METHODS

The results from Table 5 on IJB-B and IJB-C are even more valuable than those of HQ Quality because they comprise both high and low-quality datasets and have a substantial number of testing images, allowing for comprehensive model generalization assessment. The results of SMAFace and NSMAFace were expected. Due to the lower image quality of MS1MV2, NSMAFace should perform better than SMAFace sometimes. In contrast, both MS1MV3 and WebFace4M are high-quality datasets with less noise, suggesting that hard samples should be prioritized, i.e., SMAFace should outperform NSMAFace. Observations confirmed these expectations, indicating that the quality of the training dataset alters our training approach. On the IJB-B dataset, the model trained on Web-Face4M achieved SoTA performance at TAR@FAR=$0.01\%, 0.0001\%$, surpassing all other methods trained through WebFace4M.

Regarding TinyFace, SMAFace still holds a slight advantage due to its emphasis on hard samples. Though its persuasiveness is less than that of IJB-B and IJB-C, SoTA performance was achieved on TinyFace's Closed-set rank-5 retrieval and AVG, surpassing the accuracies of all other methods trained using all training datasets.

Table 3: The accuracy comparison of models trained on ResNet50, tested on the CALFW and VGG2-FP dataset.

| Method | Train Data | $\alpha$ | CALFW | VGG2-FP |
|---|---|---|---|---|
| AdaFace (Reproduce) | MS1MV2 | - | 96.05 | 95.16 |
| SMAFace | MS1MV2 | 0.50 | 96.13 | 95.36 |
| | | **1.00** | 96.08 | **95.38** |
| SMAFace$^*$ | | 2.00 | 96.13 | 95.10 |
| NSMAFace$^*$ | MS1MV2 | $-1.00$ | 96.12 | 95.10 |
| | | **$-3.00$** | **96.22** | 95.18 |
| | | $-5.00$ | 96.08 | 95.10 |

Table 4: Ablation of hard threshold parameter $p_0$, with the ResNet50 backbone trained by MS1MV2. For IJBC datasets, TAR@FAR=$0.01\%$, $0.001\%$ are reported. For TinyFace, closed-set rank retrieval (Rank-1, Rank-5 and Rank-20) is reported.

| Method | $p_0$ | TinyFace | | | IJBC | | | |
| | | Rank-1 | Rank-5 | Rank-20 | IJB-B | | IJB-C | |
| | | | | | $0.001\%$ | $0.01\%$ | $0.001\%$ | $0.01\%$ |
| AdaFace (Reproduce) | - | 65.37 | **69.31** | 71.62 | 87.20 | 94.77 | 93.73 | 96.12 |
| SMAFace | 0.40 | 65.50 | 69.31 | 71.62 | 85.94 | 94.63 | 93.29 | 96.05 |
| | 0.50 | 65.37 | 69.07 | 71.73 | **88.91** | 94.67 | 93.67 | 96.13 |
| | 0.60 | 64.89 | 68.94 | **71.81** | 87.99 | **94.85** | **93.83** | 96.20 |
| | 0.70 | **65.61** | 69.05 | 71.57 | 87.17 | 94.64 | 93.75 | **96.22** |
| | 0.80 | 64.94 | 68.70 | 71.24 | 86.36 | 94.42 | 93.20 | 96.02 |

Table 5: A performance comparison of recent methods ArcFace(Deng et al., 2019a), URL(Shi et al., 2020), CurricularFace(Huang et al., 2020b), MagFace(Meng et al., 2021), DAM-CurricularFace(Liu et al., 2021), PASS(Dhar et al., 2021), 3D-BERL(He et al., 2022), IDEA-Net(Low & Beng-Jin Teoh, 2022), AdaFace(Kim et al., 2022) and VPL-ArcFace(Deng et al., 2021) on IJB-B, IJB-C and Tiny-Face datasets with the ResNet100 backbone. For IJB-B and IJB-C, TAR@FAR=$0.01\%$, $0.0001\%$ is reported. Closed-set rank retrieval (Rank-1 and Rank-5) is used for TinyFace. The first table contains training sets all from MS1MV2, the second table consists of training sets from MS1MV3, and the third one utilizes WebFace4M. All SMAFace variants have $\alpha = 1$, SMAFace* has $\alpha = 2$ and NSMAFace* variants all have $\alpha = -2$.

| Method | IJB-B | | IJB-C | | TinyFace | | |
| | $0.0001\%$ | $0.01\%$ | $0.0001\%$ | $0.01\%$ | Rank-1 | Rank-5 | AVG |
| ArcFace | 38.28 | 94.25 | 89.06 | 96.03 | | | |
| URL | | | | 96.60 | 63.89 | 68.67 | 66.28 |
| CurricularFace | | 94.80 | | 96.10 | 63.68 | 67.65 | 65.67 |
| MagFace | 40.91 | 94.33 | **89.26** | 95.81 | | | |
| DAM-CurricularFace | | 95.12 | | 96.20 | | | |
| PASS | | | | 94.60 | | | |
| 3D-BERL | 45.77 | 94.98 | 88.45 | 96.20 | | | |
| IDEA-Net | | | | | 66.13 | | |
| AdaFace (Reproduce) | 45.37 | 95.47 | 87.94 | 96.76 | 68.11 | **71.67** | **69.89** |
| **SMAFace** | 45.64 | 95.24 | 86.77 | 96.66 | **68.19** | 71.27 | 69.73 |
| **NSMAFace*** | **46.28** | **95.60** | 86.52 | **96.80** | 67.84 | 71.22 | 69.53 |
| VPL-ArcFace | | 95.56 | | 96.76 | | | |
| AdaFace (Reproduce) | 43.26 | 95.84 | 91.42 | 97.08 | 68.03 | **71.08** | 69.56 |
| **SMAFace** | 41.22 | **95.89** | **91.81** | **97.18** | **68.40** | 71.06 | **69.73** |
| **NSMAFace*** | **43.41** | 95.78 | 89.59 | 97.11 | 67.97 | 70.90 | 69.44 |
| ArcFace | | 95.75 | | 97.16 | 71.11 | 74.38 | 72.75 |
| MagFace | 40.91 | 94.51 | 90.24 | 95.97 | | | |
| AdaFace (Reproduce) | 49.81 | 96.03 | 90.45 | **97.31** | **72.29** | 74.73 | 73.51 |
| **SMAFace** | **49.82** | 95.84 | 91.39 | 97.30 | 72.08 | **75.11** | **73.60** |
| **SMAFace*** | 44.29 | **96.06** | 91.53 | **97.31** | 72.10 | 74.73 | 73.42 |
| **NSMAFace*** | 48.63 | 95.81 | **91.99** | 97.23 | 71.97 | 74.41 | 73.19 |

