# OpenReview forum: "SMAFace: Sample Mining Guided Adaptive Loss for Face Recognition"
_ICLR.cc/2024/Conference — ICLR 2024 Conference Withdrawn Submission_

### Official Review · Reviewer_kbyJ · 2023-10-24

**Soundness:** 2 fair
**Presentation:** 2 fair
**Contribution:** 1 poor
**Rating:** 3
**Confidence:** 5

**Summary:**

In this paper, the authors propose their method, namely SMAFace, to leverage the technique of sample mining and margin-softmax for learning face recognition network. In the experiments conducted by the authors, the proposed method shows certain advantage in the results.

**Strengths:**

The paper is straightforward and easy to follow.

**Weaknesses:**

-	The motivation of this work is ambiguous. In Abstract, the authors introduce the challenge of noisy training data, but the proposed method is developed for improving model training on hard samples, rather than noisy sample learning.
-	The definition of ``information-dense’’ sample is confusing.
-	Why easy sample belongs to so-called ``information-dense’’ along with hard sample? Why easy sample present distinct feature as hard sample does?
-	The definition of $\gamma$ is not given. Also does the ``GST’’.
-	The index of the equation between Eqn3 and Eqn4 is not given. (Let us note it as Eqn3.5) Eqn3.5 is also confusing, including two similar sub equations. The condition is not given.
-	The color notion (red and blue) in Tab2 is not defined.

**Questions:**

Please refer to the weakness.

---

> ### Author Response · Authors · 2023-11-13
> **Addressing Reviewer Feedback and Enhancing Clarity in Our Paper**
>
> Firstly, I would like to express my gratitude for taking the time to thoroughly review our paper and for providing valuable feedback. Below are our responses to each of the shortcomings you have identified:
>
> Regarding the issue of unclear motivation:
> We greatly appreciate your feedback on this matter. It is true that we overlooked the distinction between noisy samples and hard samples, which is an issue we plan to address in our next paper. The challenge lies in how to differentiate between hard samples and noisy samples, as both have low probabilities and are difficult to distinguish from each other. Our method takes a somewhat straightforward approach in handling them; when the probability is extremely low, our "st" value is less than 0, effectively treating such samples as noisy samples, thus minimizing their impact on our training errors. For cases where the probability is low but not extremely so, we give them the attention they deserve, categorizing them as hard samples. We admit that our abstract did not clearly convey this distinction. We mentioned the difficulties faced by previous models in handling low-quality and noisy samples, with low-quality samples often being mistaken for hard samples rather than noise. In the process of revising the paper, we will re-examine the abstract and introduction sections to ensure a clearer expression of our research motivation.
> Thank you for your valuable feedback, and we are committed to improving the clarity of our paper in light of these comments.
>
> Concerning the definition of "information-dense" samples:
> Your input is highly valuable. Indeed, our explanation of "information-dense" samples may not have been clear. We conducted the analysis primarily using IJB-B and IJB-C datasets due to their large number of test images. Through experiments involving these datasets, it becomes evident that, for training sets like MS1MV3 with high quality and minimal noise, emphasizing challenging samples often yields better results. This is because the training set has less noise, making challenging samples the "information-dense" ones in this context. Conversely, in training sets with more noise and inadequate cleaning, such as MS1MV2 or WebFace42M, prioritizing simple samples often leads to better performance. In this case, simple samples are referred to as "information-dense" samples. This is why we categorize both simple and challenging samples as "information-dense" samples.
> Thank you for highlighting this issue, and we are committed to enhancing the clarity of our definitions and explanations in the revised paper.
>
> Regarding the issue of undefined terms and equations:
> The $\gamma$ is a modulating factor. GST is the gradient scaling parameter. Regarding Equation 3.5, we indeed failed to specify when to use SMFace and when to use SMAFace. Even in the appendix, we did not clarify this, which is a critical oversight, and we appreciate your pointing it out. In fact, due to computational and time constraints, we only had the opportunity to experiment with SMFace in the past month. Its name should have been more appropriately labeled as SM-CosFace, as it will be used to demonstrate how our method improves recognition accuracy when applied to CosFace. If needed, we can provide the .ckpt model files trained with SMFace and training records on wandb. Ultimately, for model training, we will directly choose SMAFace because it incorporates an adaptive margin. We will add clear definitions for $\gamma$ and GST in the paper and ensure that the index between Equation 3 and Equation 4 is clearly indicated. We will also provide clarification for Equation 3.5 with explicit condition descriptions.
> Thank you for bringing these issues to our attention, and we are committed to addressing them in the revised paper.
>
> Regarding the undefined color concepts in the tables:
> I apologize for the confusion that arose during your reading. The red data in the table represents the state of the art, while the blue data corresponds to the second-highest accuracy ranking. We will provide clear definitions of the color concepts, red and blue, in Table 2 in the paper to ensure that readers can understand their significance.
>
> Thank you for bringing this issue to our attention, and we are committed to providing the necessary clarifications in the revised paper.

---

### Official Review · Reviewer_tEeY · 2023-10-28

**Soundness:** 2 fair
**Presentation:** 2 fair
**Contribution:** 2 fair
**Rating:** 3
**Confidence:** 5

**Summary:**

This paper incorporates sample mining into margin-based methods to improve performance in dealing with low-quality images. It prioritizes information-dense samples and employs a probability-driven mining strategy to enhance robustness and adaptability. Experimental results show that SMAFace outperforms state-of-the-art methods on some datasets.

**Strengths:**

SMAFace introduces an adaptive training method for hard-negative mining. Compared to the counterpart without this module, it improves the accuracy of face recognition.

**Weaknesses:**

1.The quality of writing and layout of the paper need improvement.
2.By integrating marginal softmax, there are too many hyperparameters. It seems more like a parameter-tuning technique, lacking novelty.
3.The results show only a slight improvement, without demonstrating significant effects.

**Questions:**

None

---

> ### Author Response · Authors · 2023-11-13
> **Response to Reviewer Comments and Acknowledgement of Feedback**
>
> Firstly, we would like to express our gratitude for your diligent review of our submitted paper and for your valuable feedback. We greatly appreciate your time and professional insights. Below are our responses to your review comments:
>
> Regarding the issues of paper quality and layout:
> We have acknowledged your suggestions, and we are committed to enhancing the writing quality and layout of the paper. We will carefully review and revise the paper to ensure it achieves a higher standard in these aspects.
>
> Concerning the excessive number of hyperparameters:
> You mentioned the issue of introducing too many hyperparameters with the integration of the margin-based softmax method. We will provide clearer descriptions of these hyperparameters in the paper to reduce reader confusion and ensure that their selection is well-justified. We want to clarify that, for our baseline methods, we have not altered their hyperparameters. The numerous hyperparameters primarily pertain to the parameters we introduced, as they are novel, and therefore, we need to determine their optimal values through ablation experiments.
>
> Regarding the minor performance improvements:
> You noted that the results show only slight improvements. We acknowledge that performance improvements on some datasets may not be very significant, but we believe this is a valuable research direction, especially when dealing with low-quality images. We will express this more clearly in the paper and provide additional experiments to demonstrate the effectiveness of our approach. In the past month, we have applied Sample Mining to CosFace, resulting in SM-CosFace, which achieved higher accuracy than the original method. This demonstrates the value of our approach.
>
> We sincerely appreciate your review comments, which are highly beneficial for improving our work. We will resubmit the revised paper, and we hope you will reconsider our work. Once again, thank you for your valuable feedback.

---

### Official Review · Reviewer_fauv · 2023-10-31

**Soundness:** 3 good
**Presentation:** 1 poor
**Contribution:** 2 fair
**Rating:** 3
**Confidence:** 4

**Summary:**

The paper proposed a new SMAFace face recognition algorithm that adaptively integrates sample mining into the margin-based loss function, which is claimed to handle low-quality face images well. The paper also proposed a so called scaling term to analyze face recognition method. The proposed method was evaluated 4 public datasets and compared favorably with baseline methods.

**Strengths:**

The paper presented an in-depth discussion about adaptive loss function for low-quality face images. Then the paper proposed to integrate hard sample mining to dynamically adjust weight coefficient based on the probability of the correct class, which appears an effective way to further improve face recognition performance for low-quality face images.

**Weaknesses:**

First of all, the paper shall be self-contained within 9 pages with/without the Appendix. However, this paper appears to take the Appendixes as indispensable components, with many references to the Appendixes in the main text.  If the main text is the just the outline and proof and detailed explanation are in the Appendixes, this may violate the 9-page limit in some sense.

For example, if the scaling term is the 2nd contribution of the paper, why only explains its details in Appendix?

“Regarding the adaptive margin function, we define it as” … “For a detailed explanation, please refer to Appendix A”, why not explain the key contribution of adaptive margin function in the main text?

The paper is not well-written and hard to follow. To list a few examples: “which as been presented in the Appendix” in the 4th paragraph in the introduction; “when a sample is more hard”; “Combining it with fields beyond FR promises encouraging results”.

Important reference missing:
DeepFace: closing the gap to human-level performance in face verification, CVPR 2014.
Deep learning face representation by joint identification-verification, NIPS 2014

**Questions:**

Other than the loss function, SMAFace pretty much followed Deng et al (2019a), e.g., using the backbone ResNet, what the performance would be if using the new ViT network?

---

> ### Author Response · Authors · 2023-11-13
> **Enhancing Clarity and Content Localization in SMAFace: A Response to Reviewer Feedback**
>
> Dear Reviewer,
>
> Thank you for taking the time to review our submission titled "SMAFace: Adaptive Sample Mining for Face Recognition." We greatly appreciate your constructive feedback, and we have carefully considered your comments in our revision. Below is our response to your feedback:
>
> Self-Containment and Appendix Usage: We apologize for any potential confusion regarding the use of the Appendix. We understand your concerns about staying within the 9-page limit and have taken them into account. We have indeed relocated the proof of content to the appendix, placing the core conclusions and essential content in the main text. The content in the appendix is non-essential for readers, and we will remove many unnecessary references accordingly. In our revised version, we will ensure that essential content is moved from the Appendix to the main text to maintain self-containment.
>
> Explanation of Contributions: We appreciate your suggestion to provide a more comprehensive explanation of the scaling term in the main text for improved clarity. Thank you for raising concerns about the adaptive margin function. The reason for not including it in the main text is that it is not the primary contribution of this paper; rather, it functions as a tool, akin to a controlled variable in an experiment. Over the past month, we have configured it as a non-adaptive CosFace and conducted corresponding experiments, specifically the SM-CosFace experiments, to demonstrate that the inclusion of Sample Mining indeed contributes to improved recognition rates.
>
> Improving Clarity: We acknowledge your feedback regarding the clarity of the paper and commit to working on improving the overall readability and coherence of the manuscript. We will carefully proofread the document and address any grammatical or language issues.
>
> Missing References: We apologize for omitting important references, including "DeepFace: closing the gap to human-level performance in face verification, CVPR 2014" and "Deep learning face representation by joint identification-verification, NIPS 2014." We will include these references in our revised manuscript.
>
> Exploring Alternative Architectures: We appreciate your suggestion to explore the use of the new ViT network as an alternative to ResNet. ViT is undoubtedly the latest and most popular network framework, and using it for training yields highly anticipated results. However, due to time constraints and limited computational resources, we did not consider it in our paper. We plan to supplement our code with ViT implementation in the future open-source release, making it available for readers with sufficient computational resources to reference and use.
>
> We would like to express our gratitude for your valuable feedback, and we are committed to addressing the concerns you have raised to enhance the quality and clarity of our paper. We hope that our revisions will meet your expectations and lead to a more favorable evaluation of our work.
>
> Thank you once again for your time and expertise in reviewing our submission.

---

### Comment · Area_Chair_DZ9k · 2023-12-05
**Final Update**

Dear Reviewers,

Please take this chance to carefully read the rebuttal from the authors and make any final changes if necessary.

Please also respond to the authors that you have read their rebuttal, and give feedback whether their rebuttal have addressed your concerns.

Thank you,

AC